# Genomic analysis of T Cell receptors reveals lynch syndrome specific immune signatures

Nan Deng[1,13], Fahriye Duzagac [1,13], Ana M. Bolivar[1,13], Laura Reyes-Uribe[1], Melissa W. Taggart[2], Selvi Thirumurthi[3], Luigi Ricciardiello[3], Patrick M. Lynch[3], Y. Nancy You [4], Scott Kopetz [5], Paul Scheet [6], Gregory A. Lizee [7,8], Alexandre Reuben [9], Fatima Marin[10], Marta Pineda [10], Krishna M. Sinha[1], Ajay Bansal[11,12], Gabriel Capella[10] & Eduardo Vilar [1,5] ✉

Lynch Syndrome (LS) provides the perfect context to understand DNA mismatch repair deficient carcinogenesis, which is characterized by neoplastic lesions with high rates of shared neoantigens eliciting adaptive immunity through T cell receptor (TCR) recognition. However, the TCR landscape in LS carriers remains unexplored. Here, we perform TCR sequencing of 277 blood samples from LS cancer survivors, previvors, and controls, as well as matching colorectal cancers and pre-cancers. We show that up to 41% of the most expanded TCRβs from colorectal neoplasms are detectable in the blood of LS carriers, while showing minimal expansion in controls. In addition, we develop and validate a classification model that distinguishes LS carriers from controls using circulating TCRβs signatures associated with LS independent of the-cancer history and with cancer-free LS previvors. Together, our findings characterize circulating and tissue TCRβs associated with LS, thus representing a step toward identifying blood-based TCR biomarkers for immune surveillance.

Lynch Syndrome (LS) is a genetic condition estimated to affect more than one million individuals in the United States[1,2]. LS carriers inherit a heterozygous germline pathogenic variant in one of four DNA mismatch repair (MMR) genes (*MLH1*, *MSH2*, *MSH6*, and *PMS2*), which predisposes them to develop different types of cancers, starting at a young age[3]. LS is the primary cause of hereditary colorectal cancer (CRC), conferring a 50-80% lifetime risk as well as a 40–60% for endometrial cancer (EC), and enhances the risk to develop multiple other tumor types[4]. MMR deficiency occurs when cells acquire a second somatic hit in the alternate allele of the gene that harbors the germline pathogenic alteration. This event leads to the accumulation of hundreds of somatic mutations, primarily insertion-deletions (indels) in microsatellite (MS) loci[5].

Therefore, malignant MMR-deficient (MMRd) cells produce high levels of tumor-specific neoantigens (neoAgs), derived primarily from the MS indels, which trigger an anti-cancer immune response via the

[1]Department of Clinical Cancer Prevention, The University of Texas MD Anderson Cancer Center, Houston, TX, USA. [2]Department of Pathology, The University of Texas MD Anderson Cancer Center, Houston, TX, USA. [3]Department of Gastroenterology, Hepatology and Nutrition, The University of Texas MD Anderson Cancer Center, Houston, TX, USA. [4]Department of Colorectal Surgery, The University of Texas MD Anderson Cancer Center, Houston, TX, USA. [5]Department of GI Medical Oncology, The University of Texas MD Anderson Cancer Center, Houston, TX, USA. [6]Department of Epidemiology, The University of Texas MD Anderson Cancer Center, Houston, TX, USA. [7]Department of Melanoma Medical Oncology, The University of Texas MD Anderson Cancer Center, Houston, TX, USA. [8]Department of Immunology, The University of Texas MD Anderson Cancer Center, Houston, TX, USA. [9]Department of Thoracic/Head and Neck Medical Oncology, The University of Texas MD Anderson Cancer Center, Houston, TX, USA. [10]Department of Hereditary Cancer Program, Catalan Institute of Oncology - ICO, Hereditary Cancer Group, ONCOBELL Program, Institut d'Investigació Biomèdica de Bellvitge - IDIBELL, Ciber Oncología (CIBERONC) - Instituto de Salud Carlos III, L'Hospitalet de Llobregat, Barcelona, Spain. [11]Division of Gastroenterology and Hepatology, The University of Kansas Medical Center, Kansas City, KS, USA. [12]The University of Kansas Cancer Center, Kansas City, KS, USA. [13]These authors contributed equally: Nan Deng, Fahriye Duzagac, Ana M. Bolivar. ✉e-mail: EVilar@mdanderson.org

major histocompatibility complex (MHC)[6]. Yet, subsequent activation of immune checkpoint regulators enables malignant cells to evade this immune response and continue to proliferate uncontrollably[7]. Many of these neoAgs are widely shared among LS carriers, as several coding MS (cMS) loci have high mutation propensity, thus resulting in identical mutations occurring at common loci in tumors from different LS individuals[8]. Similar to viral antigens, these neoAgs are highly immunogenic, eliciting T cell activation and immune memory that enables rapid responses upon re-exposure[9]. Following the elimination of the neoAgs, proliferated T-cells undergo contraction and continue to exist as a smaller population of memory T-cells[10], which may enter the bloodstream and circulate within the body[11]. As a result, the diversity and abundance of peripheral blood TCR clonotypes reflect the individual's history of antigen exposure

Public TCRs, defined as clonotypes shared across individuals due to recurring antigen exposure, offer a window into the shared immunological memory[12]. Mapping public TCRs has diagnostic potential[13,14]. The 'TCR-Antigen Map' initiative aimed to catalogue TCRs binding clinically relevant antigens across diseases. As proof of concept, it identified cytomegalovirus (CMV)-specific TCRs[15], an approach that ultimately led to FDA approval of the T-Detect COVID test[16]. Given the unique context of LS as a well-defined, high-risk MMRd cancer model, the extensive sharing of neoAgs among LS individuals across disease stages, and the limited characterization of LS-associated TCR repertoires, there remains a critical gap in understanding how shared, antigen-experienced TCR clonotypes are represented systemically.

Here, we show that analysis of T cell receptor landscapes from 277 peripheral blood mononuclear cell (PBMC) samples from 102 LS MMRd cancer survivors, 130 LS carriers without history of cancer (LS previvors), and 45 controls without LS or history of cancer. Additionally, we performed TCR-sequencing (TCRseq) in colorectal tissues of 3 cancers and 11 pre-cancers matching their PBMC. Together, these findings reveal that circulating cancer-associated TCRs can be identified in the peripheral blood of LS carriers and provide insights into the immune-biology mechanisms influencing cancer susceptibility in LS individuals.

## Results
### Demographic and clinical characteristics of the LS patient cohort

We analyzed 277 PBMC or whole blood samples from 102 LS survivors, 130 LS previvors, and 45 controls collected at MD Anderson Cancer Center (MDACC), Kansas University, and the Catalan Institute of Oncology (ICO-IDIBELL, Spain, Table 1). LS survivors had a mean age of 54 years (range 26–82) and were predominantly female (61%), whereas LS previvors had a mean age of 44 years (range 18–85) with 71% females; controls had a mean age of 59 years (range 28–76) with balanced sex distribution (49% female, 51% male). Most LS survivors carried pathogenic variants in *MSH2* (42%), followed by *MLH1* (26%), *MSH6* (21%), and *PMS2* (11%). LS previvors showed a similar pattern: *MSH2* (34%), *MLH1* (23%), *MSH6* (23%), and *PMS2* (20%). Among survivors, 72 had colorectal cancer (CRC) and 24 had endometrial cancer (EC), with fewer cases of pancreatic, urothelial, gastric, small bowel, ovarian, and prostate cancers. We also analyzed 14 matching colorectal lesions from LS carriers in our PBMC/whole-blood cohort, including 11 tubular adenomas and three adenocarcinomas (Table S1). Among the 14 lesions, four were MMRd, and three were MMR-proficient (MMRp) by IHC; two lesions had indeterminate IHC due to ambiguous staining. The remaining five samples had been exhausted for TCRseq, leaving MMR status unavailable (Table S2).

### Clonality and diversity of circulating TCRs

We assessed diversity of circulating TCRβ chains using Hill numbers, which estimate the effective number of clonotypes with equal abundance; higher values indicate greater diversity. Our analysis revealed

that LS survivors exhibited significantly ($P = 0.0006$, two-tailed Mann-Whitney test) lower diversity levels compared to LS previvors (median 97.38 *vs*. 191.4, respectively) and compared to controls ($P = 0.076$, two-tailed Mann-Whitney test, median controls: 160.2, Fig. 1A). This trend persisted when using the Simpson clonality index (SCI), which assesses both clonotype numbers and their degree of clonal expansion; an SCI value closer to 1 indicates lower diversity, whereas a value closer to 0 indicates higher diversity. LS survivors exhibited significantly ($P = 0.0002$, two-tailed Mann-Whitney test) higher clonality in their TCR repertoires than LS previvors (median 0.03500 *vs*. 0.02200, respectively), and a marginally significant difference ($P = 0.094$, two-tailed Mann-Whitney test) was also observed between LS Survivors and controls (median 0.02660, Fig. 1B). These differences likely reflect a higher frequency of large, hyperexpanded TCRβ clonotypes in LS survivors, whereas LS previvors show greater abundance of rare, minimally expanded clonotype (Fig. 1C).

We observed weak positive correlations between the age of individuals and the level of clonality among survivors (Spearman's correlation, $R = 0.31$), previvors ($R = 0.31$), and controls ($R = 0.22$, Figure S1A). Only among females, a significant difference in SCI between LS survivors and LS previvors was observed (median 0.03655 for LS survivors *vs*. 0.01900 for LS previvors, $P = 0.00022$, two-tailed Mann-Whitney test, Figure S1B). Similarly, regardless of the originating institution of the sample, LS survivors exhibited more clonal repertoires than LS previvors and controls (Figure S1C). No significant differences were detected in the diversity of the TCRβ repertoires among LS survivors and previvors when analyzed based on the mutated MMR genes (Figure S1D). The effects of disease status and age were further confirmed using a generalized linear model (GLM) analysis, which considered disease status, institution, gender, mutant MMR gene, and HLAs as covariates. Ultimately, only age and disease status were found to be statistically associated with the SCI (Tables S3 and S4).

### Public TCRβ repertoires and similarity within cancer status groups

We first focused on circulating TCRβs shared among individuals in our PBMC/blood samples, as these were of particular interest to us. We considered a TCRβ to be *"shared"* or *"public"* if it was present in at least two individuals with an identical amino acid sequence match of the CDR3β, Vβ, and Jβ segments. The majority of these public TCRβs (a total of 1,232,112, Jaccard Index of 0.617) were shared by both LS survivors and previvors (Fig. 2A and B). To better understand the differences in TCR sharing between the three groups, we then used a rank–size frequency distribution analysis[17,18]. The data showed that only 3 public TCRs (0.000539%) in the control cohort were shared by 50% of controls. In contrast, 69 public TCRs (0.00454%) in LS previvors and 13 public TCRs (0.00197%) in LS survivors were shared by 50% of samples in each group (Fig. 2C). Additionally, only 4,058 public TCRs (0.73%) were shared by the control cohort, while 25,731 TCRs (1.69%) in LS previvors and 12,231 TCRs (1.85%) in LS survivors were shared by 10% of the samples in each group, respectively. The power exponent α value of the power-law distribution equation was 0.307 for controls, 0.368 for previvors, and 0.405 for survivors, thus suggesting that the previvor and survivor groups share more public TCRs than the controls ($P = 1\times10^{-16}$ for both previvor and survivor groups compared to controls in the F test, Fig. 2C).

### TCRβ repertoire overlap between colorectal lesions and peripheral blood

Our primary goal was to identify a circulating TCRβ signature reflecting neoAg exposure in colorectal tissue, thus we investigated whether TCRβ sequences from CRC and pre-cancers were detectable in matched blood. We conducted bulk TCR-sequencing on 14 colorectal lesions from LS carriers collected during their routine surveillance colonoscopy (Table S1). The TCRβ repertoire from each tissue sample

**Table 1 | Summary of patient demographics and characteristics for PBMC and whole-blood samples**

| Characteristic | N = 277 | % |
|---|---|---|
| **Cancer status** | | |
| Survivor | 102 | 37 |
| Previvor | 130 | 47 |
| Control | 45 | 16 |
| **Age** | | |
| Survivors | | |
| Mean ± Stdv | 54 ± 13 | N/A |
| Min - Max | 26–82 | N/A |
| Previvors | | |
| Mean ± Stdv | 44 ± 13 | N/A |
| Min - Max | 18–85 | N/A |
| Controls | | |
| Mean ± Stdv | 59 ± 10 | N/A |
| Min - Max | 28 - 76 | N/A |
| **Sex (P = 0.02, Chi-squared test)** | | |
| Survivors | | |
| Female | 62 | 61 |
| Male | 40 | 39 |
| Previvors | | |
| Female | 92 | 71 |
| Male | 38 | 29 |
| Controls | | |
| Female | 22 | 49 |
| Male | 23 | 51 |
| **Race (P = 0.3, Chi-squared test)** | | |
| Survivors | | |
| White or Caucasian | 88 | 86 |
| Black or African American | 4 | 4 |
| Asian | 3 | 3 |
| Native Hawaiian or Other Pacific Islander | 0 | 0 |
| Other | 6 | 6 |
| Unknown | 1 | 1 |
| Previvors | | |
| White or Caucasian | 119 | 92 |
| Black or African American | 3 | 2 |
| Asian | 1 | 1 |
| Native Hawaiian or Other Pacific Islander | 1 | 1 |
| Other | 5 | 3 |
| Unknown | 1 | 1 |
| Controls | | |
| White or Caucasian | 28 | 62 |
| Black or African American | 4 | 9 |
| Asian | 1 | 2 |
| Native Hawaiian or Other Pacific Islander | 0 | 0 |
| Other | 3 | 7 |
| Unknown | 9 | 20 |
| **MMR gene (P = 0.2, Chi-squared test)** | | |
| Survivors | | |
| MLH1 | 27 | 26 |
| MSH2 | 43 | 42 |
| MSH6 | 21 | 21 |
| PMS2 | 11 | 11 |
| Previvors | | |
| MLH1 | 30 | 23 |

**Table 1 (continued) | Summary of patient demographics and characteristics for PBMC and whole-blood samples**

| Characteristic | N = 277 | % |
|---|---|---|
| MSH2 | 44 | 34 |
| MSH6 | 30 | 23 |
| PMS2 | 26 | 20 |
| **Institution** | | |
| MDACC | 180 | 65 |
| Kansas University | 53 | 19 |
| ICO-IDIBELL | 44 | 16 |
| **LS-associated cancer types in LS Survivors*** | | |
| Colorectal | 72 | N/A |
| Endometrial | 24 | N/A |
| Urothelial | 5 | N/A |
| Pancreas | 4 | N/A |
| Gastric | 3 | N/A |
| Small Bowel | 1 | N/A |
| Ovarian | 3 | N/A |
| Prostate | 3 | N/A |
| **History of other non-LS cancer*** | | |
| Lung | 1 | N/A |
| Lymphoma | 2 | N/A |
| Basal cell carcinoma | 7 | N/A |
| Squamous cell carcinoma | 9 | N/A |
| Sarcoma | 2 | N/A |
| Breast | 10 | N/A |
| Sebaceous carcinoma | 3 | N/A |
| Thyroid | 5 | N/A |
| Melanoma | 2 | N/A |

*One patient can have more than one cancer

N/A not applicable.

was used to analyze the overlap between the top 100 most expanded clonotypes in each lesion and the matched patient's blood samples. Figure 3A and B show the top 100 most expanded TCRβs in cancer and tubular adenomas. A total of 41 and 28 of the most expanded TCRβs were also identified in the matched blood, respectively, even when the blood was collected at later timepoints after the biopsy (Fig. 3C and Table S2). Some of these overlapping TCRβs showed medium and large expansion in the blood and, most importantly, some were identified as public TCRβs shared by different LS individuals (Fig. 3A and B, red font). Overall, on average, 18% of the top 100 expanded TCRs in CRCs and pre-cancers were present in blood collected at the same time as the lesions. This overlap slightly decreases to an average of 14% when blood samples are taken 1-10 years after detecting the lesions (Figure S1E). Figure S2 summarizes the overall overlap of the most expanded blood-derived TCRβs between cancer and tubular adenoma samples. Moreover, we observed that 3 to 25% of the top 100 most expanded tissue-infiltrating TCRβs from each colorectal lesion were also present in the circulating public TCRβ repertoire of our patient cohort (Fig. 3C). Therefore, our findings suggest that these TCRs may be targeting neoAgs derived from the colorectal tissue and T-cell responses are detectable both in the lesion microenvironment and in circulation.

**Annotation of public TCRβ repertoire to disease-specific TCRs**
To assess the specificity of circulating public TCRβs, we annotated them against reference repertoires with known antigen specificities (Fig. 4A). For this, we used the McPAS-TCR database, which curates TCR sequences linked to diverse diseases and antigens[19]. From this

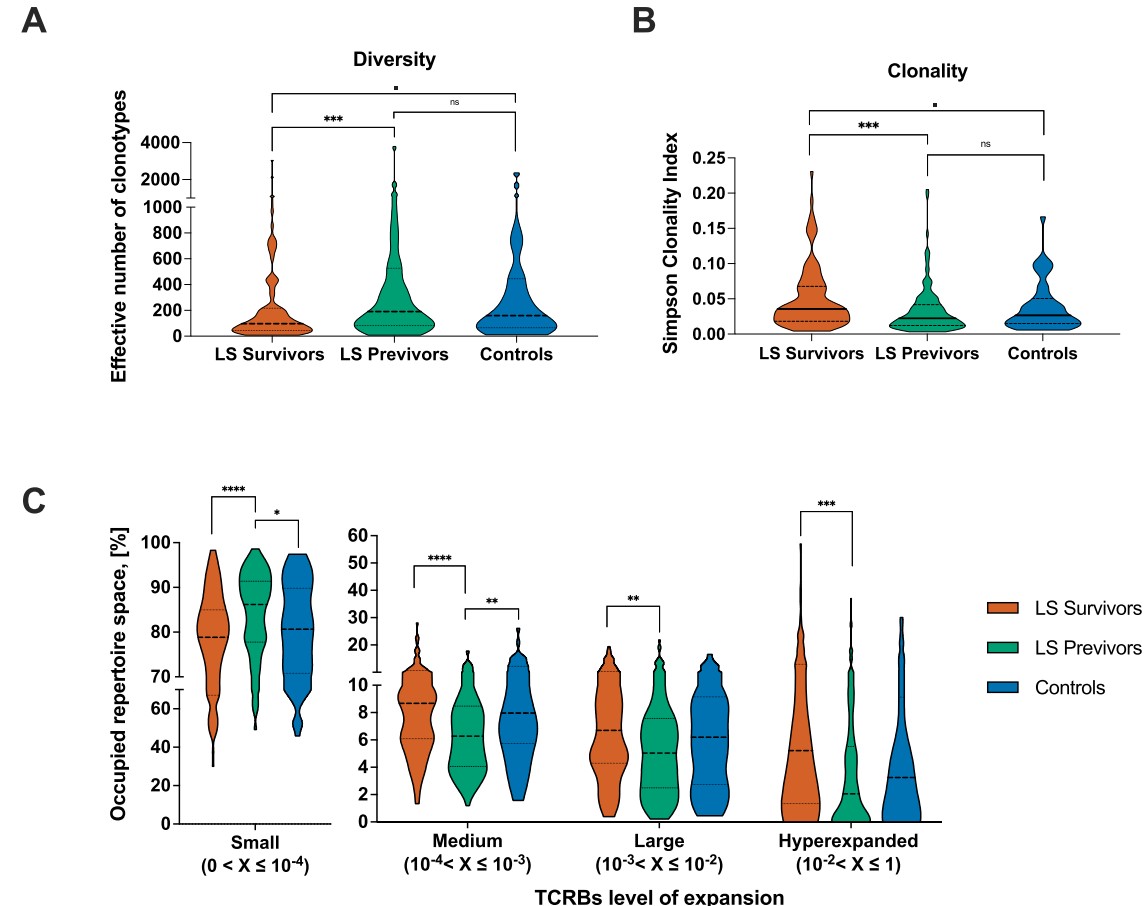

**Fig. 1 | Diversity and clonality of TCRβ repertoires in peripheral blood according to LS-associated cancer status. A** Comparison of effective number of clonotypes (Hill numbers) among survivors (N = 102), previors (N = 130) and controls (N = 45, Two-tailed Mann-Whitney test; controls vs LS Previors: *P* value = 0.5; controls vs LS Survivors: *P* = 0.08; LS Previors and LS Survivors: *P* value = 6e-4; **B** Comparison of clonality (Simpson clonality index) among survivors (N = 102), previors (N = 130) and controls (N = 45, Two-tailed Mann-Whitney test; controls vs LS Previors: *P* value = 0.3; controls vs LS Survivors: *P* value = 0.08; LS Previors and LS Survivors: *P* value = 6e-4); **C** Comparison of the percentage of repertoire space occupied by unique TCRβs with specific levels of expansion, among survivors (N = 102), previors (N = 130) and controls (N = 45, Mann-Whitney two-tailed test, Hyperexpanded (10-2 < X ≤ 1), LS Previors vs Controls: *P* value =

0.41668; Hyperexpanded (10-2 < X ≤ 1), LS Survivors vs Controls: *P* value = 0.08083; Hyperexpanded (10-2 < X ≤ 1), LS Survivors vs LS Previors: *P* value = 0.00022; Large (10-3 < X ≤ 10-2), LS Previors vs Controls: *P* value = 0.265; Large (10-3 < X ≤ 10-2), LS Survivors vs Controls: *P* value = 0.26809; Large (10-3 < X ≤ 10-2), LS Survivors vs LS Previors: *P* value = 0.00251; Medium (10-4 < X ≤ 10-3), LS Previors vs Controls: *P*-value = 0.00453; Medium (10-4 < X ≤ 10-3), LS Survivors vs Controls: *P* value = 0.9648; Medium (10-4 < X ≤ 10-3), LS Survivors vs LS Previors: *P* value = 1e-05; Small (0 < X ≤ 10-4), LS Previors vs Controls: *P* value = 0.03937; Small (0 < X ≤ 10-4), LS Survivors vs Controls: *P* value = 0.26089; Small (0 < X ≤ 10-4), LS Survivors vs LS Previors: *P* value = 1e-05; ****P value < 0.0001 ***P value < 0.001; *P value < 0.05; • P value < 0.1; ns, non-significant). Source data are provided as a Source Data file.

database, we extracted two pools: one comprising 24,884 TCRs specific to viral pathogens and another with 2,113 TCRs specific to autoimmune conditions. We also compiled a pool of 14,208 MMRd CRC–specific TCRβs by integrating sequencing data from nine MMRd CRCs previously reported in the literature[20]. Interestingly, more TCRβs were annotated to the MMRd CRC pool of tissue-resident TCRβs compared to the viral pathogens pool in survivors (15.5% *vs.* 5.3%, $P = 1.56 \times 10^{-13}$, Chi-Square test), previors (18.2% *vs.* 6.1%, $P = 5.75 \times 10^{-19}$, Chi-Square test), and controls (9.4% *vs.* 2.7%, $P = 2.3 \times 10^{-6}$, Chi-Square test), even though the viral pathogens-specific TCR pool contains nearly twice the number of TCRs compared to the pool of TCRs in MMRd CRC (Fig. 4B). Additionally, when we examined the expansion of TCRβs associated with MMRd CRC in the blood and correlated it with their expansion in CRC tissue samples, LS survivors and previors exhibit significantly higher levels of expansion in the tissue compared to the blood samples ($P < 0.0001$, two-tailed Mann-Whitney test, Fig. 4C). Overall, these results indicate that LS survivors harbor MMRd CRC–specific public TCRβs with medium to large expansion in peripheral blood, whereas LS previors and controls show medium to small expansion.

## Circulating TCRβs to distinguish LS carriers from the general population

We next evaluated whether a distinct circulating public TCRβ signature could differentiate LS carriers from controls. The average expansion of this signature of TCRβs in the blood of a given sample is expected to predict whether the sample comes from an LS carrier or a healthy control (LS vs control status).

To identify the TCRβ signature associated with LS, we tested several methods including clustering approaches like GLIPH, Levenshtein distance, and TCRdist, which treat similar TCRs as the same, and an exact match method (Supplementary Data 1). However, the more complex methodologies did not show clear improvement in performance over the exact match approach. Therefore, to streamline the interpretability and simplicity of our method, we used the exact match classifier. To identify the best set of exact match signatures, we considered the following hyperparameters: 1. TCR region: the most optimal region to use among CDR3aa only, CDR3aa plus J segments (CDR3aa+J), CDR3aa plus V segments (CDR3aa+V), or CDR3aa plus both J and V segments (CDR3aa+J + V); 2. Positive presence ratio cutoff: TCRβ clones must be present in a certain percentage of individuals in

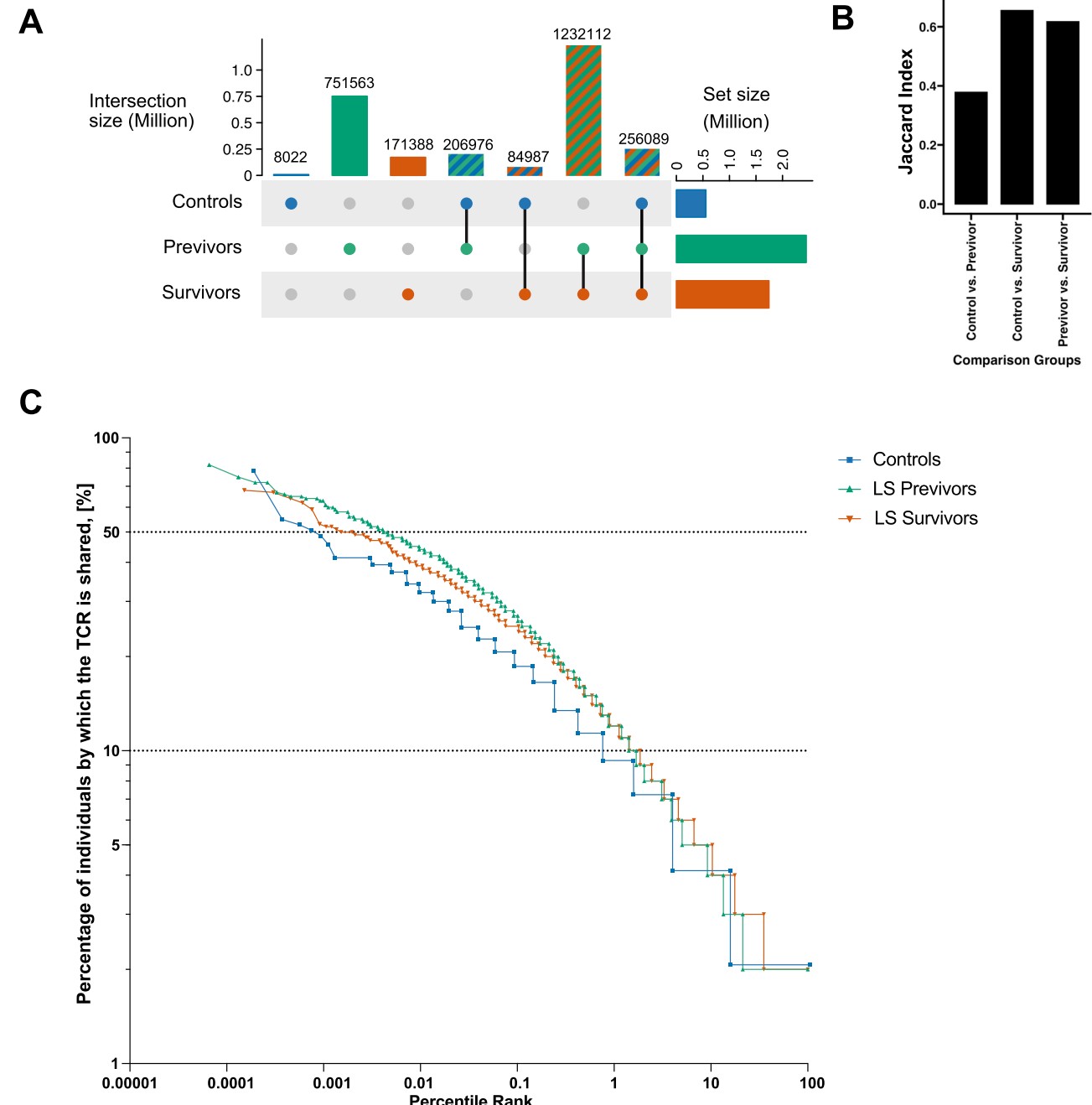

**Fig. 2 | Public TCRβ repertoire and similarity within LS cancer status groups.**
**A** Upset plot showing the overlap of public TCRβs among LS survivors (N = 102), LS previvors (N = 130), and controls (N = 45); **B** Jaccard Index between different groups; **C** Rank–size distribution of TCRs in the three groups (control, previvor, and survivor). The frequency was normalized to the percentage of the total samples in each group. The rank was normalized to the percentile of the rank. The axes are in a log10 scale. Control vs LS previvor: $P$ value < $2 \times 10^{-16}$; Control vs LS survivor: $P$ value < $2 \times 10^{-16}$; LS previvor vs LS survivor: Not Significant in two tailed F test. Source data are provided as a Source Data file.

the positive cohort; 3. Fisher's test cutoff: a particular TCRβ clone must be significantly enriched (occurrence) in LS individuals compared to controls by passing Fisher's exact test at a given threshold; 4. Wilcoxon test cutoff: a particular TCRβ clone must show a significant increase (overall level of expansion) in LS individuals compared to controls, as determined by the Wilcoxon test at a given threshold. Then, we randomly assigned 80% of the samples to a training set (Supplementary Data 2) to select the best hyperparameters, and 20% to a validation set. To compensate for the imbalance in the numbers of negative controls, we also included 197 healthy controls from a public dataset[21]

(Supplementary Data 3). We then used a Leave-One-Out Cross-Validation (LOOCV) strategy to test combinations of the four hyperparameters in the training set: TCR region (CDR3aa, CDR3aa+J, CDR3aa +V, CDR3aa+J + V), positive presence ratio cutoff (10%, 50%, 75%), Fisher's test cutoff ($10^{-1}$, $10^{-2}$, $10^{-4}$, $10^{-10}$, $10^{-15}$), and Wilcoxon test cutoff ($10^{-1}$, $10^{-2}$, $10^{-4}$, $10^{-10}$, $10^{-15}$, Fig. 5A). The AUROC (Area Under the Receiver Operating Characteristic curve) was used as the performance metric.

We found that using only the CDR3aa region to represent the TCR, requiring TCRs to appear in at least 10% of LS subjects, and applying

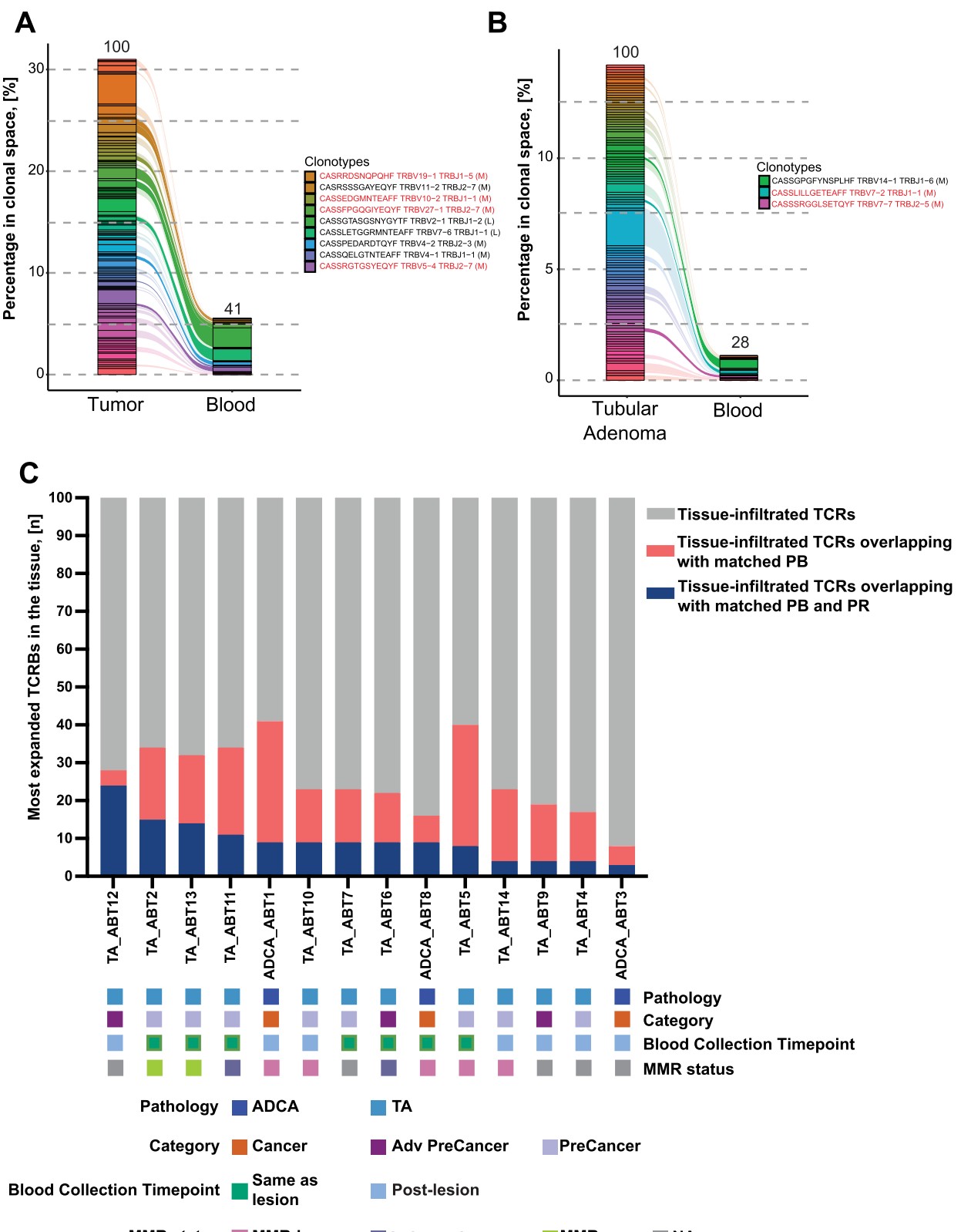

both Fisher's and Wilcoxon test cutoffs at $10^{-10}$ yielded the best LOOCV performance on the training set (Supplementary Data 1). Using this set of hyperparameters, we identified 13,340 TCRβs associated with the LS phenotype compared to controls (Supplementary Data 4 and Figure S3). These TCRβs exhibited higher expansion and incidence among LS previvors and survivors compared to controls (Fig. 5B, C). To

rule out overfitting, we applied this TCR signature set to the remaining 20% validation set and achieved an AUROC of 0.930, which was slightly lower but comparable to the performance on the training set (AUROC = 0.942, Fig. 5D). Overall, these findings highlight the capability of TCRβ signatures to effectively distinguish LS carriers from the general population.

**Fig. 3 | TCRβ overlap between colorectal lesions and matched peripheral blood.** **A** overlap of the top 100 most expanded TCRβs of an LS adenocarcinoma (sample ADCA_ABT1, MMR deficient) with its matched blood sample TCRβ repertoire; **B** Overlap of the top 100 most expanded TCRβs of an LS tubular adenoma (sample TA_ABT12, MMR status unavailable) with its matched-patient blood sample TCRβ repertoire. In **A**, **B** the red font indicates that the TCRβ is part of the public TCRβ repertoire from our sample cohort. The letters M and L (next to clonotype) stand respectively for medium and large expansion of TCRβ in the blood; **C** Summary of

the overlap between the top 100 most expanded tissue-infiltrating TCRβs, the matched blood samples (peripheral blood, PB), and the public TCRβ repertoire from our cohort for all tissue samples (public repertoire, PR). The bottom panel displays pathological characteristics of each sample and the collection timepoint of the blood sample compared to the timepoint in which the lesion biopsy was obtained, as covariate bars. Adv PreCancer Advanced Pre-cancer, TA Tubular Adenoma, ADCA Adenocarcinoma (Stage I-III). Source data are provided as a Source Data file.

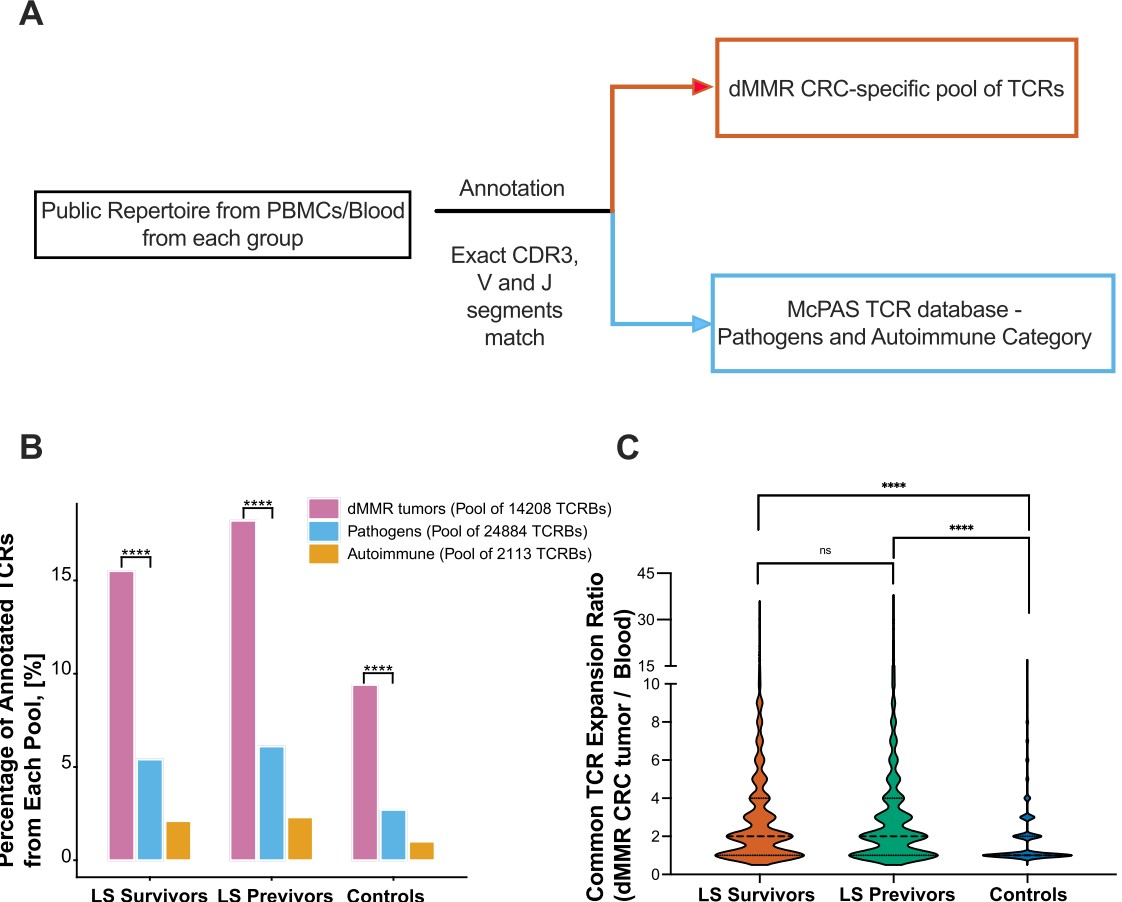

**Fig. 4 | Annotation of public TCRβs to TCRs with known specificity. A** Schematic of the annotation process; **B** Percentage of TCRβs from each disease-specific pool that annotate to LS TCR cohort from survivors, previvors, and controls (Two-tailed Mann-Whitney test, tissue-resident TCRβs vs. the viral pathogens pool in survivors: $P$ value $= 1.56 \times 10^{-13}$; in previvors: $P$ value $= 5.75 \times 10^{-19}$; in controls: $P$ value $= 2.31 \times 10^{-6}$); **C** Differences between LS Survivors, LS Previvors and controls in terms

of the TCRs expansion ratio in the tumor over their expansion in the blood (Two-tailed Mann-Whitney test, LS Previvors vs Controls: $P$ value $= <2e-16$; LS Survivors vs Controls: $P$ value $= <2e-16$; LS Survivors vs LS Previvors: $P$ value $= 0.857$; ****$P$ value $< 0.0001$ ***$P$ value $< 0.001$; *$P$ value $< 0.05$; • $P$ value $< 0.1$; ns non-significant) Source data are provided as a Source Data file.

## Circulating TCRβs classification to distinguish LS previvors from the general population

We next asked whether TCRβs could identify LS carriers without cancer (previvors). Using the same classification approach, we tested whether the TCRβ repertoire distinguished Controls from Previvors within the LS cohort (Fig. 6A). When we applied the same strategy to identify the signatures in LS Previvors vs Control, the best LOOCV classification accuracy measured by the AUROC was achieved using the same hyperparameter set as the LS carrier vs Control classifier (TCR region=CDR3aa, Positive Presence Ratio = 10%, Fisher's test cutoff = $10^{-10}$, Wilcoxon test cutoff = $10^{-10}$, Supplementary Data 5). Under these parameters, we identified 16,248 TCRβ sequences associated with the LS previvor phenotype (Supplementary Data 6 and Figure S4), resulting from increased expansion and incidence in LS

previvors compared to controls (Fig. 6B, C). In the training dataset, the classifier distinguished LS previvors from controls with high performance (AUROC = 0.953), which was also confirmed in the validation dataset (AUROC = 0.925, Fig. 6D). We also evaluated the classifier after excluding the public dataset and obtained a reasonable LOOCV performance for control vs LS carriers (AUROC = 0.835, Figure S5A), and control vs previvors (AUROC = 0.836, Figure S5B). These results highlight the presence of a TCRβ signature capable of effectively distinguishing LS previvors from controls.

## Circulating TCRβs classification to distinguish LS survivors from LS previvors

We then investigated whether LS survivors could be distinguished from LS previvors using a classifier built from TCRβ repertoires

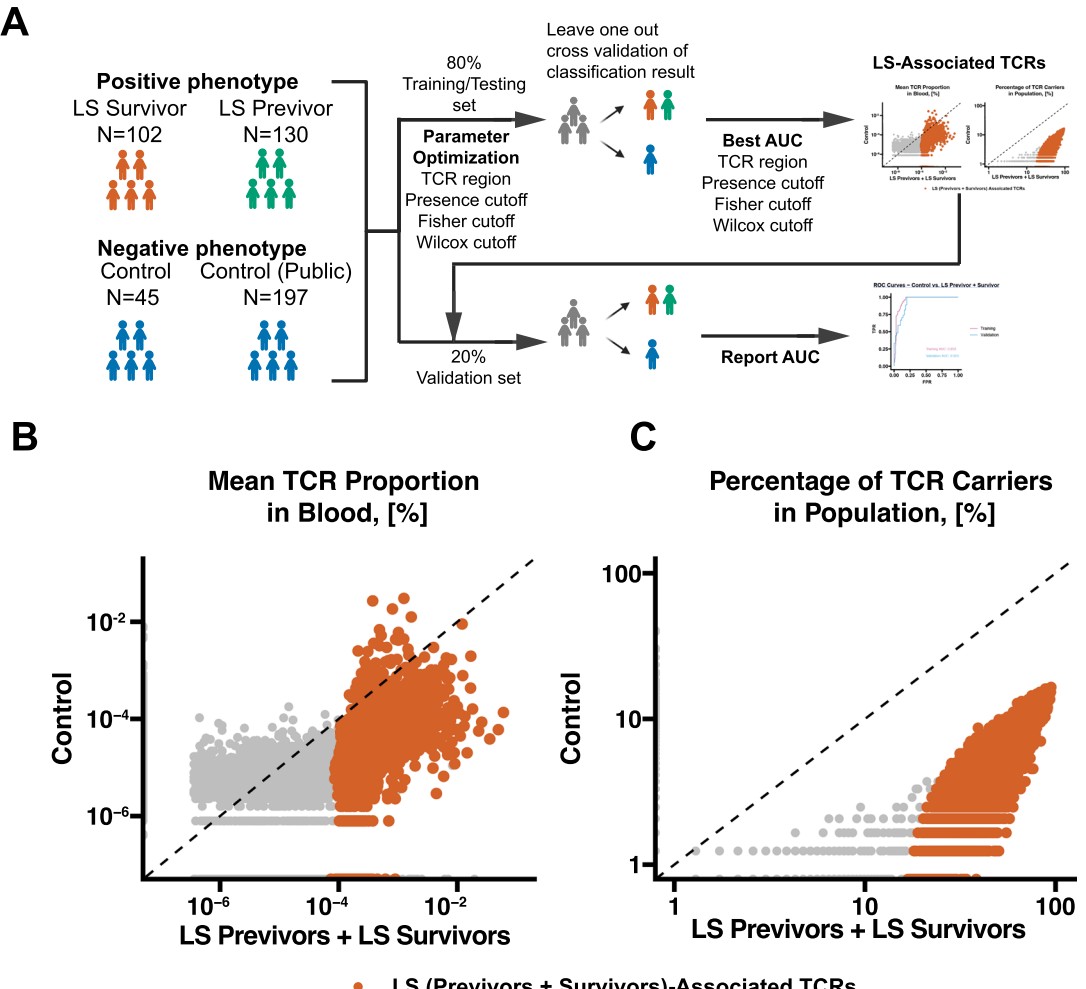

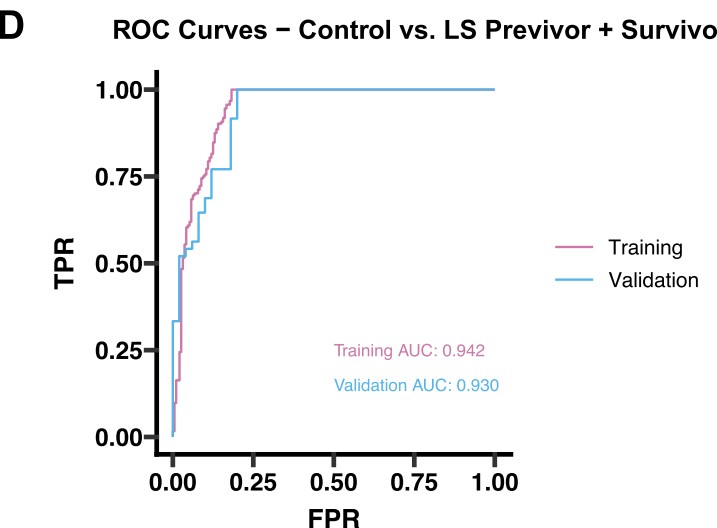

**● LS (Previors + Survivors)-Associated TCRs**

**Fig. 5 | Classification of PBMCs/Blood samples as LS vs. Control status, based on circulating TCR signatures. A** Schematic of the experimental design; **B** Level of expansion of public TCRβs used for the classifier in LS carriers (Previor + Survivor) vs. controls. Red dots indicate Lynch carriers (Previor + Survivor)-associated TCRβ signatures, and grey dots indicate all other public TCRβs; **C** Incidence ratio of the number of individuals with LS carriers or controls in which the public TCRβ signatures used for the classifier were present. Red dots indicate Lynch-associated TCRβ signatures and grey dots indicate all other public TCRβs; **D** Area under the receiver operating characteristic curves indicating the performance of the LS-associated TCRβ signatures in the classification of samples as either an LS individual or a control in the training and validation datasets. TPR, true positive rate; FPR, false positive rate. Source data are provided as a Source Data file.

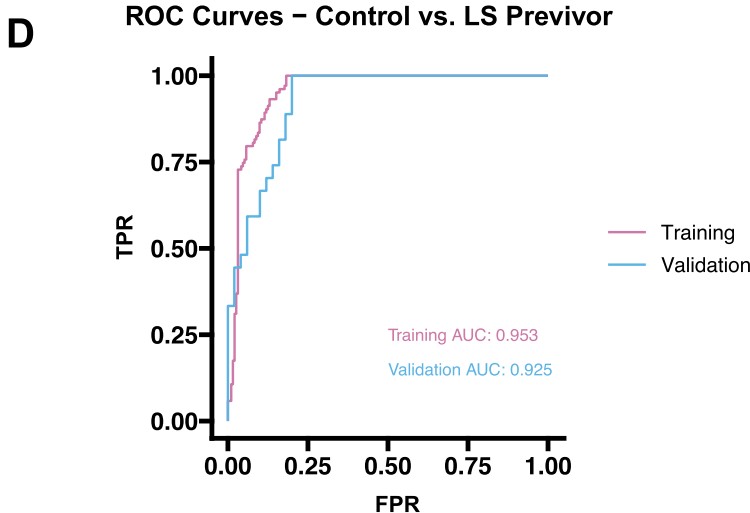

**Fig. 6 | Classification of LS PBMCs/Blood samples as Control vs. Previor status, based on circulating TCR signatures. A** Schematic of the study design; **B** Level of expansion of public TCRβs used for the classifier in LS previors vs. controls. Red dots indicate Lynch carriers previor-associated TCRβ signatures, and grey dots indicate all other public TCRβs; **C** Incidence ratio of the number of individuals with LS previors or controls in which the public TCRβ signatures used for the classifier were present. Red dots indicate LS previors associated TCRβ signatures, and grey dots indicate all other public TCRβs; **D** Area under the receiver operating characteristic curves indicating the performance of the LS previor-associated TCRβ signatures in the classification of samples as either a control or an LS previor in the training and validation datasets. TPR true positive rate; FPR false positive rate. Source data are provided as a Source Data file.

(Fig. 7A). We focused on the CDR3 aa region when selecting the signature for the previvor vs survivor comparison. The highest classification accuracy measured by AUROC was achieved using a classifier that required the TCRs to be present in >10% of survivors and an unadjusted *P* value cutoff of 0.1 for both Fisher's exact test and the Wilcoxon test (Supplementary Data 7). With these parameters, we identified a total of 581 expanded TCRβ clones associated with the phenotype observed in LS survivors (Supplementary Data 8 and Figure S6). These TCRs demonstrated a greater clonal expansion and higher incidence in survivors compared to previvors (Fig. 7C). Notably, the previvor vs survivor signature is integrated by 581 TCRs, which is a substantially smaller number when compare the other two groups: control vs LS carriers (13,340 TCR clones) and control vs previvors (16,248 TCR clones), despite being defined under a less stringent statistical cut-off (FDR = 0.1 vs $10^{-10}$). The predictive model derived from this signature achieved only moderate discriminative performance (AUROC = 0.732; Fig. 7D).

TCRβ signatures distinguishing controls from LS carriers (previvors and survivors combined) showed substantial overlap with the signatures distinguishing controls from LS previvors alone (12,697 out of 16,248; Jaccard Index = 0.76, Fig. 7E), while only 7 out of 574 signatures (Jaccard Index = 0.0005) distinguishing LS survivors from previvors were found in the signatures distinguishing the controls from all LS carriers (previvors and survivors together). We also observed that the mean pairwise Jaccard index was significantly higher between previvors and survivors ($0.0128 \pm 0.00553$) compared with control vs previvor ($0.0101 \pm 0.00410$) and control vs survivor ($0.00948 \pm 0.00370$) comparisons (two-tailed Wilcoxon test, $P < 1 \times 10^{-22}$, Fig S7A and S7B). In addition, substantially fewer TCRs clones showed significant differences at different *P* value cutoffs in abundance using Wilcoxon test (Fig S7C) or incidence using Fisher test (Fig S7D) between previvors and survivors. Together, these observations indicate a higher overall similarity of TCR clones between previvors and survivors, which likely accounts for the limited classifier performance to distinguish them. The similarity between TCR repertoires suggests that LS previvors may have already been exposed to many of the same antigens before tumor development occurs in LS survivors.

### In vitro validation of TCR-seq analysis

We next validated these findings using in vitro immunological assays. We leveraged annotated public TCRβs classified as viral pathogen–specific in the McPAS-TCR database (Figs. 4A, B), and selected for validation a subset previously reported as EBV-specific for the BMLF1 epitope (GLCTLVAML). An EBV peptide-tetramer complex was used to isolate antigen-specific CD8+ T-cells by flow cytometry for single-cell TCR sequencing (Figure S8A). EBV-tetramer–specific CD8+ T cells comprised 3.56% of total cells (Figure S8B) and were compared with EBV-annotated public TCRs in McPAS-TCR. A total of 47 TCRβs overlapped between the EBV-tetramer-specific CD8+ T-cells and our circulating public TCRβ cohort (Supplementary Data 9 and Figure S8C). Of these, five TCRβs were annotated as EBV-BMLF1–specific in McPAS-TCR, thus validating the accuracy of our annotation findings (highlighted by purple arrows, Figure S8D).

### Validation of the LS neoAg-specificity from the circulating Lynch-associated TCR signature

After refining our experimental approach using an EBV-tetramer, we then validated our TCRβ LS-phenotype signatures for two highly recurrent neoAg peptides, RNF43 (TQLARFFPI) from two LS carriers (IDs S_ABDL106 and ID S_ABDL165), and MSH3 (FLLALWECSL) from another LS carrier (ID S_ABDL166, Fig. 8A) using dextramer staining followed by paired scRNA and scTCR sequencing. We had previously demonstrated that these two neoAg are immunogenic *in vitro*[22]. After staining, 2.1% of cells were successfully isolated as activated RNF43-

specific CD3+ T-cells from a LS carrier (ID S_ABDL106, Fig. 8B and Figure S9A, top). This population included both CD4+ and CD8+ T-cells. We then repeated the RNF43 neoAg staining with the addition of CD8 gating in a second LS carrier (S_ABDL165, Figure S9A, bottom), and also included a second neoAg, MSH3, from another LS carrier (S_ABDL166, Figure S9B). This refined approach revealed that 10.6% of CD8+ T-cells were RNF43-specific in a LS carrier (S_ABDL165, Fig. 8B and Figure S9A, bottom), while 6.38% of CD8+ T-cells were MSH3-specific in another LS carrier (S_ABDL166, Fig. 8B and Figure S9B). Figure S9C displays the corresponding UMAP projections of CD3+, CD4+, and CD8+ T-cell populations derived from these single-cell datasets, confirming consistent clustering and annotation across samples (gating strategy shown in Figure S10). Notably, upon examining the intersection between the circulating LS-associated TCRβ signature, which was derived from the classifier distinguishing all LS carriers (survivors and previvors) from controls, and the RNF43-dextramer-positive CD3+ and CD8+ T-cells, we identified a total of 80 TCRβ sequences that overlapped between the two sets originating from a T-cell exhibiting CD3+ and CD8+ phenotype (Supplementary Data 10 and Fig. 8C). These shared TCRβs represent ~6.4% of the total unique CDRβ sequences (n = 1249) obtained from the two RNF43-specific samples. From the MSH3-specific CD8+ T-cell population, we identified 20 TCRs whose CDR3β sequences overlapped with the circulating LS-associated TCRβ signature, thus representing ~5.2% of the total unique CDR3β sequences (n = 388) in that sample (Supplementary Data 10 and Fig. 8D). These results provide direct evidence that a subset of circulating LS-associated TCRβ signatures identified in this study are neoAg-specific and detectable at the single-cell level, thereby validating their relevance to immune surveillance in LS carriers.

### Validation of the CRC-specificity from the circulating LS-associated TCR signature

Finally, we examined the convergence between the LS-associated signatures derived from the LS carriers (previvors and survivors) vs controls (Fig. 5B, C) and the MMRd CRC-resident TCR pool, which was previously compiled for annotation (Figs. 4A, B). This MMRd CRC-resident pool was generated by combining TCRβ repertoires from nine publicly available MMRd CRC[20] and two additional MMRd ADCAs obtained from our tissue cohort at MD Anderson. This analysis revealed 783 overlapping TCRβs between these two datasets (Fig. 8E and Supplementary Data 11). Of note, 8 of these overlapping TCRβs displayed notable expansion (Frequency>0.5%) within the tumor microenvironment of the annotated MMRd CRC tissue samples (Fig. 8F). The observed overlap between LS-associated CDR3β sequences and those identified in RNF43+ scRNA-seq and MMRd CRC data supports the notion that frequent exposure to shared neoAgs, such as RNF43-derived neoepitopes, drives the convergence of TCRβ repertoire and is responsible for the immune surveillance in LS. Taken together, these observations suggest that some TCRβs associated with LS demonstrate specificity towards MMRd CRC and neoAgs presented by these malignancies.

### Discussion

Here, we show that a large-scale characterization of the TCR repertoire in LS carriers reveals that immune responses within colorectal lesions are not fully confined to the local microenvironment. Public TCRβ clonotypes detected in colorectal lesions are also detectable in the circulation. Importantly, we developed a classification model that uses a set of LS-associated public TCRβs to distinguish LS carriers from controls with high accuracy.

Blood-derived TCRs are established biomarkers of prognosis, diagnosis, and treatment monitoring in multiple cancers. For example, PBMC repertoires predicted therapy response and survival in head and neck cancer[23], and HER2+ breast cancer patients displayed more significantly expanded circulating CD8+ clones than HER2− tumors[24].

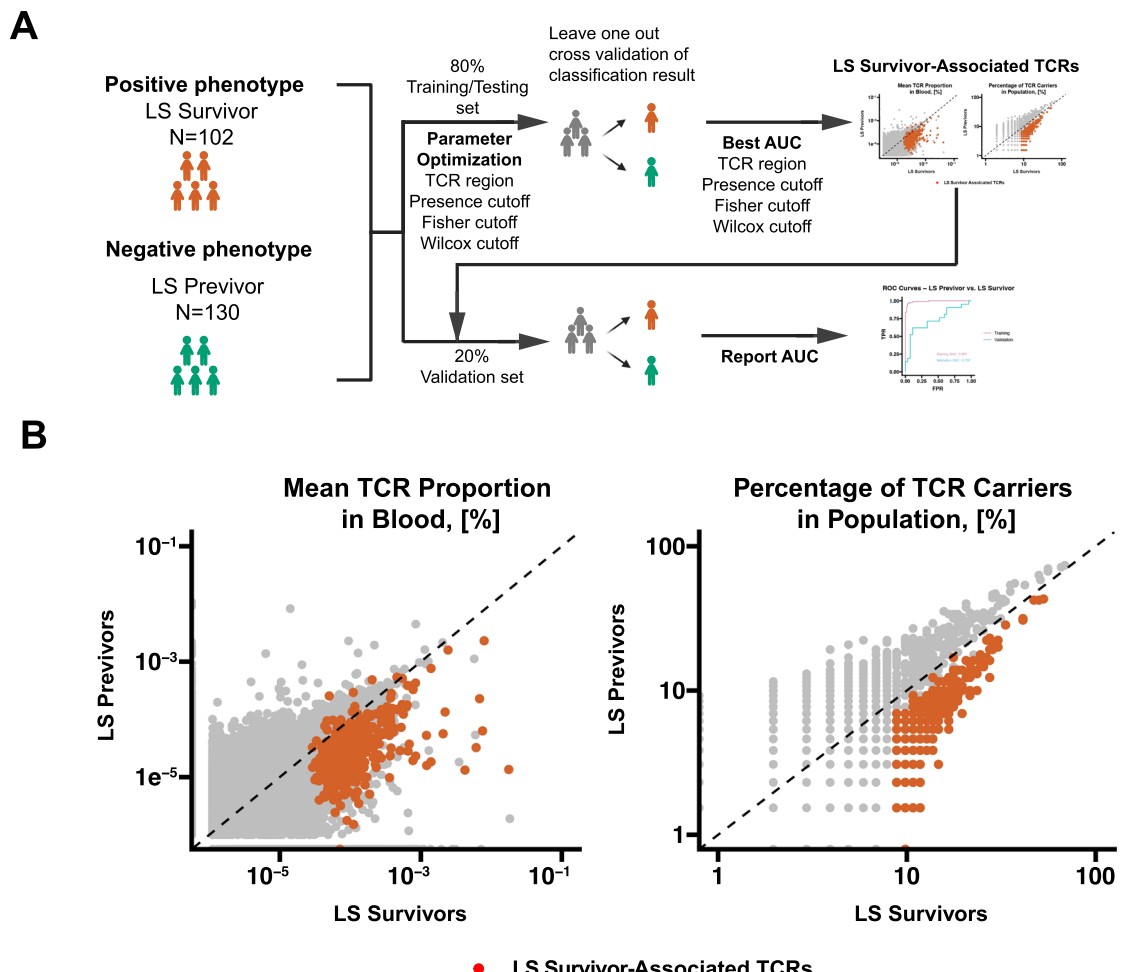

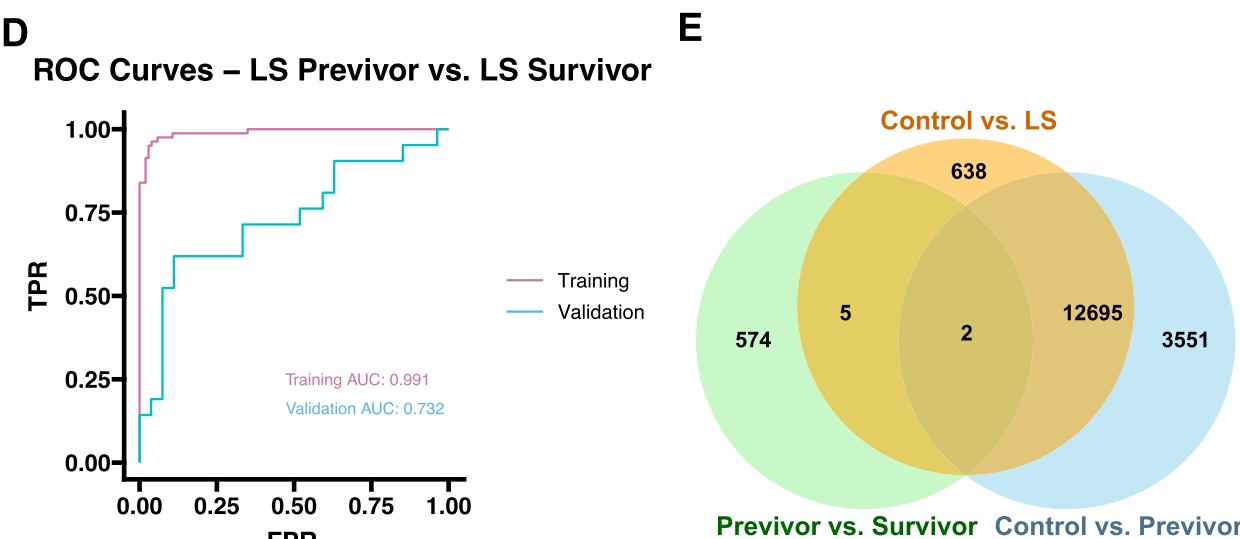

**Fig. 7 | Classification of PBMCs/Blood samples as LS previvors vs. survivor status, based on circulating TCR signatures.** **A** Schematic of the experimental design; **B** Level of expansion of public TCRβs, used for the classifier, in LS previvors vs. LS survivors. Red dots indicate LS survivor-associated TCRβ signatures and grey dots indicate all other public TCRβs; **C** Incidence ratio of the number of individuals with LS previvors or LS survivors in which the public TCRβ signatures used for the classifier were present. Red dots indicate LS survivor-associated TCRβ signatures and grey dots indicate all other public TCRβs; **D** Area under the receiver operating characteristic curves indicating the performance of the LS survivor-associated TCRβ signatures in the classification of samples as either an LS survivor or an LS previvor in the training and validation datasets. TRP, true positive rate; FPR, false positive rate; **E** Venn diagram showing the overlap of TCR signatures from three different classifiers: the controls vs. LS carriers (previvor + survivor), controls vs. LS previvors, and LS survivors vs. LS previvors. Source data are provided as a Source Data file.

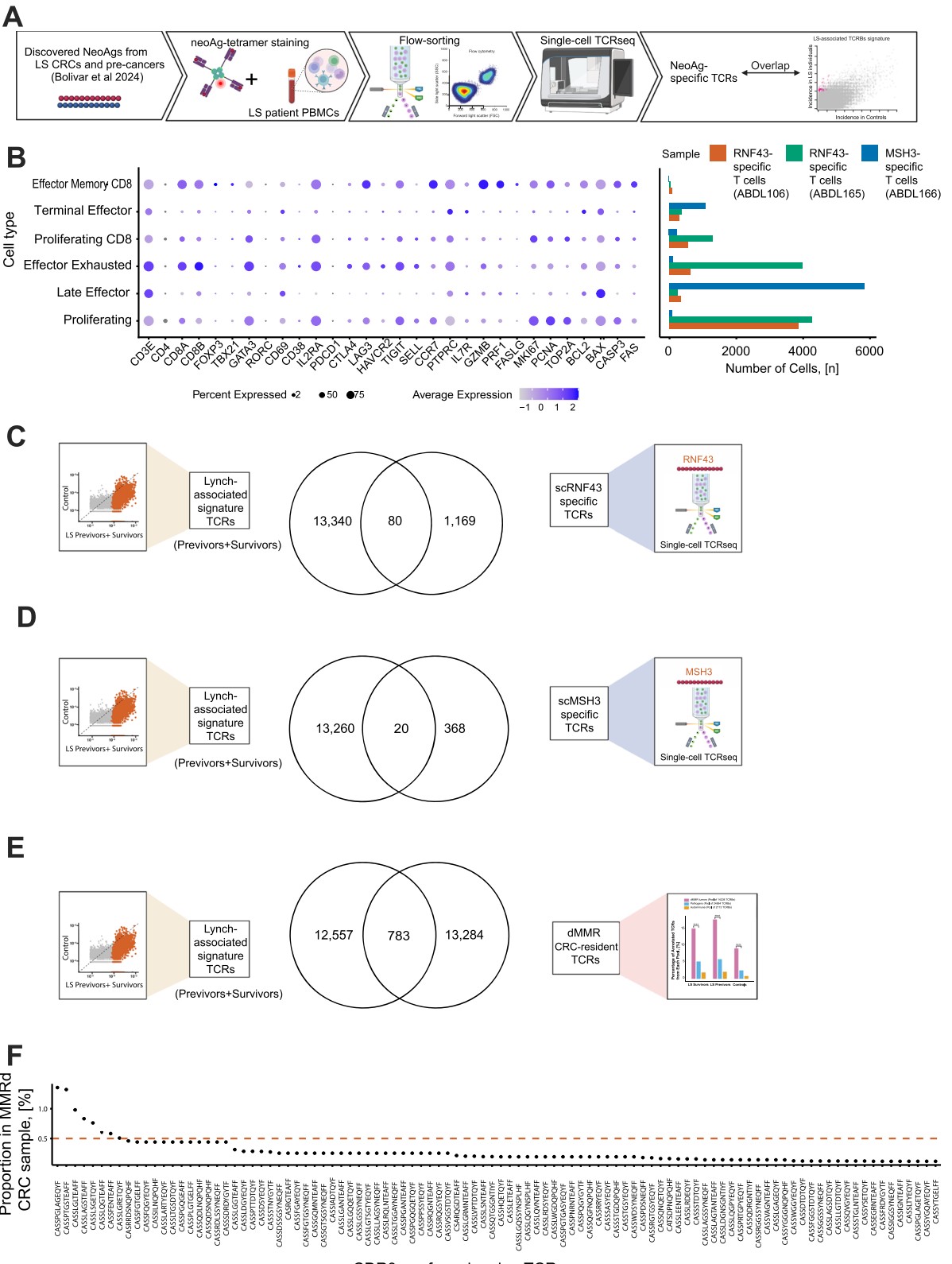

**Fig. 8 | In vitro validation of the Lynch-associated TCR signature. A** Schematic of the experimental setup; **B** Expression levels of the different T-cell types obtained from the single-cell analysis results with the RNF43 and MSH3-dextramer-specific T-cells. Bar plots show the number of RNF43- and MSH3-dextramer–specific T cells recovered from each LS carrier analyzed.; **C** Overlap of TCRs between the LS-associated signature TCRs and the RNF43-dextramer-specific CD3[+] and CD8[+] TCRs by exactly matching the CDR3 segments; **D** Overlap of TCRs between the LS-associated signature TCRs and the MSH3-dextramer-specific CD8[+] TCRs by exactly matching the CDR3 segment; **E** Overlap of TCRs between the LS-associated signature TCRs and the MMRd CRC-resident TCR pool by exactly matching the CDR3 segments; **F** Dot plot showing the proportions of the top 50 overlapping TCRs within the corresponding MMRd colorectal tumor samples, based on their CDR3 amino acid sequences (CDR3aa). Source data are provided as a Source Data file.

However, no such studies have been performed in LS individuals until now and, thus our study represents a pioneering effort.

The reduced TCR diversity observed in LS survivors compared with LS previors and cancer-free controls aligns with prior reports across multiple disease contexts. Cui et al[25] reported a gradual decline in circulating TCR diversity as cervical cancer progressed[25], and lung cancer patients also showed reduced diversity relative to healthy individuals, even before treatment[26]. In our own dataset, we also observed weak but consistent positive correlations between age and TCR clonality across all groups, including LS survivors, LS previors and controls, and confirmed age as an independent factor associated with clonality through multivariable modeling[27,28]. These findings are consistent with previous studies reporting age-associated TCR repertoire contraction and clonal expansion due to thymic involution and cumulative antigen exposure over time. Overlap of tumor-infiltrating TCRs among patients with the same cancer type has also been described in microsatellite-stable CRC, pancreatic cancer, and cervical neoplasia[25,29–31].

We observed that 18% of the top 100 expanded TCRs in colorectal cancers and pre-cancers overlapped with matched blood collected at the same time, decreasing to 14% when blood was collected >1 year after lesion detection. This limited overlap is consistent with prior studies reporting 19–25% similarity between tumor-infiltrating and circulating TCR profiles[32–34]. Our study takes a more comprehensive approach, considering the overlap of both Vβ and Jβ segments, and indicates a slightly lower percentage of overlap. Moreover, our assessment is the first to explore overlap between infiltrating TCRs from pre-cancers and circulating TCRs from matched blood samples.

Various algorithms analyze TCR repertoires to identify phenotypic associations. For instance, ImmuneML classifies phenotypes based on clone counts[35], while the immune Cell Analysis Tool (iCAT) uses presence or absence of TCRs, using Fisher's exact test[36]. Our algorithm integrates both approaches, considering clone counts and clonotype incidence, and accurately identifies signatures distinguishing controls from all LS carriers, but the performance is limited in distinguish LS previor and survivors. Immune surveillance plays an important role in editing premalignant clones prior to cancer initiation. In LS carriers MMRd leads to accumulation of indels generating a distinct but relatively narrow spectrum of mutations. In our previous work, we identified frequent MSI events in LS pre-cancers and shared neoAgs between pre-cancers and cancers. Consequently, naïve CD8+ T cells may encounter these neoAgs early in carcinogenesis, thus generating TCR repertoires that persist during tumor progression. This results in highly similar TCR signatures between LS previors and survivors, limiting the discriminatory power of TCR-based classifiers. Consistent with this, frameshift peptide–specific T cell responses have been detected in the peripheral blood of both cancer-free LS carriers and those with LS-associated CRC[37–39]. However, the dynamics of TCR evolution during carcinogenesis and the transition from previor to survivor remain unclear[22]. Early neoAg-driven TCR repertoires arising from MMR deficiency are expected to be present in both groups, whereas additional TCR expansions may occur in survivors in response to tumor-specific neoAgs. In our cohort, the number of LS survivors (n = 102) and LS previors (n = 130) is relatively limited. Therefore, the classifier distinguishing LS survivors from previors achieved an AUROC of 0.991 in the training set but dropped to 0.732 in the validation set. This discrepancy indicates overfitting, which frequently arises when machine learning models attempt to distinguish between groups that share high biological similarity. In our previous study[22], we found that LS carriers may carry a high burden of somatic mutations, corresponding neoAgs from indels present in pre-cancer samples, and neoAg-specific TCR repertoires in LS previors. These TCR clonotypes may continue to persist in circulation even after tumors develop in LS survivors, and therefore they do not clearly help distinguish between the two groups. While some of these TCR clones may expand in tumors, the dynamics of TCR repertoire evolution during the transition from normal to pre-cancer and later to cancer are not well understood. One possibility is that novel neoAgs from driver mutations arise during the pre-cancer-to-cancer transition, thus leading to additional tumor-specific TCRs. Another possibility is that the TCR repertoire remains largely unchanged, either because driver mutations are already present at the pre-cancer stage or because cancer progresses through immune escape rather than the generation of new neoAgs. In our dataset, we detected fewer differentially abundant or incident TCRs between previors and survivors (Fig. 7, S5C and S5D), which is consistent with the second possibility. Therefore, our limited ability to classify these groups is unlikely secondary to a limitation in our sample size alone and instead the shared immunogenic history that is intrinsic to LS carriers. Future exploration is warranted using larger cohorts and longitudinal sampling to clarify the dynamics of the TCR repertoire during early stages of pre-cancer and cancer initiation.

In vitro validation of RNF43- and MSH3-dextramer–positive T-cells in LS carriers revealed that circulating neoAg-specific populations are largely CD8+ in phenotype. With the addition of CD8+ gating, we confirmed CD8+ T cell responses to both RNF43 and MSH3, consistent with the recurrent and immunogenic nature of these neoAgs in LS neoplasia[8,22]. Using this approach, we identified 80 RNF43-specific and 20 MSH3-specific TCRβ sequences that overlapped with the circulating LS-associated TCRβ signature. These data provide direct evidence for the presence of circulating neoAgs-specific T cells in LS carriers and support their linkage to the LS-associated TCRβ repertoire. Although broader validation across additional neoAgs and LS individuals will be required to define the specificity of this phenomenon, large-scale isolation and profiling of neoAg-specific T cells remain technically and logistically challenging in large patient cohorts. In addition, it is important to clarify that overlap between LS survivor-associated TCR signature and RNF43 or MSH3-specific populations was not assessed due to the limited number of TCRβs in the survivors' signature and the fact that both RNF43 and MSH3 are present in both cancers and pre-cancers, thus leading to TCRs recognizing it in both LS previors and survivors. Additionally, our validation focused on colorectal lesions, which could limit applicability to other cancers. However, the immunogenic similarities between colorectal and EC[8] suggest that the identified TCRs may also predict endometrial cancer presence. Further validation in endometrial samples is necessary to confirm this hypothesis.

Moving forward, large-scale TCR repertoire profiling in LS carriers, enabled by collaborative efforts and advanced sequencing, is crucial. Identifying TCR signatures linked to specific MMRd cancers may be enhanced by stratifying patients by cancer type and restricting the "positive" group to those with active disease at sampling. Temporal fluctuations in TCRβ repertoires highlight the need for longitudinal monitoring, as immune responses vary over time. Thus, while TCR profiling holds diagnostic promise, interpretation will require integration with timepoint-specific immune data and other clinical markers. Incorporating TCRseq with germline SNPs and multi-omics could yield robust biomarkers and therapeutic targets. These TCRβ repertoires have potential as immune biomarkers; however, further validation in independent cohorts is necessary to assess their clinical utility and tumor reactivity. Detecting signatures in asymptomatic LS previors could provide a non-invasive tool for early detection, risk assessment, and personalized surveillance.

In summary, the pursuit of cancer-associated circulating TCRs in LS carriers represents a steppingstone in the development of precision prevention for this patient population and provides important insights into cancer immunology. By unraveling the intricacies of T-cell responses in LS-associated neoplasms, this research has the potential to revolutionize early cancer detection, prognostication, and therapeutic interventions in this high-risk population.

## Method

### Ethical approval

Written informed consent was obtained from all participants. The study protocol was approved by the Institutional Review Board of The University of Texas MD Anderson Cancer Center (IRB #Lab-94-032, IRB #PA12-0327, and IRB #PA13-0178), with approval from the corresponding institutional review boards at participating institutions.

### Participants and samples

Bulk TCRseq was performed from whole blood or PBMCs of 277 individuals (102 LS survivors, 130 LS previvors and 45 controls without history of cancer or LS, Supplementary Data 12) collected at The University of Texas MD Anderson Cancer Center (MDACC), The University of Kansas Cancer Center (KUCC), and the Catalan Institute of Oncology (ICO, Barcelona, Spain). For the MDACC cohort, the median follow-up duration was 62 months calculated from the date of consent to the last recorded contact; three individuals were deceased at last follow-up. For the KUCC cohort, the median follow-up was ~72 months with one deceased individual. For the ICO cohort, the median follow-up duration was 138 months; however, vital status was unknown for six individuals. Across all three cohorts, aside from the four known deceased and the six participants with unknown status, all others across the three cohorts were alive at their last follow-up (Supplementary Data 12). LS survivors were defined as LS carriers with a history of an active LS-associated cancer type. LS previvors were LS carriers with no history of any type of cancer and controls were healthy individuals at average-risk for CRC with no history of any type of cancer or family history of LS. Furthermore, we also analyzed 14 colorectal neoplasms (11 pre-cancers and 3 cancers) of the colon and rectum of LS carriers from whom we obtained PBMCs or blood. Three unstained slides (4μm thick) were obtained from each sample to perform a hematoxylin and eosin (H&E) staining and immunohistochemistry (IHC) of mismatch repair (MMR) proteins, as described previously[40]. Pathological diagnosis of tissue samples was verified by an expert gastrointestinal pathologist (M.W.T.) Tissue samples were categorized as tubular adenomas (TA) and adenocarcinomas (ADCA) based on pathological assessment and then grouped into three categories: cancers; advanced pre-cancers, which comprised TAs larger than 1 cm in size or containing villous features, or the presence of high-grade dysplasia; and pre-cancers, consisting of TAs without advanced characteristics.

### Sequencing, diversity and public TCRβ repertoire Analysis

Genomic DNA was obtained from all PBMCs, whole blood, and tissue samples using the QIAGEN DNeasy Blood & Tissue Kit (Cat. No. 69504), following the manufacturer's instructions. Library preparation of genomic DNA samples was performed by Adaptive Biotechnologies using the TCRβ ImmunoSEQ Assay. DNA samples from MDACC (N = 180) were sequenced at "Deep" resolution and all samples from KUCC and ICO (N = 97) were sequenced with Adaptive's new "Single" resolution, which is comparable to the deep resolution. Downstream analyses, including diversity and clonality indexes as well as the analysis of TCRβ homeostasis and public TCRβs were performed with the Immunarch R[41] package and the ImmunoSEQ Analyzer from Adaptive Biotechnologies. Only productive TCRβ rearrangements were considered for the analysis. To evaluate TCR diversity in each sample, we used Hill number, which is a measurement frequently used in ecology to quantify the biodiversity of ecosystems. It allows for measuring the effective number of distinct clonotypes that are equally abundant corresponding to diversity[42]. The Simpson clonality index was obtained by calculating the square root of the sum over all observed rearrangements of the square fractional abundances of each rearrangement. Productive clonality was obtained by calculating the square root of Simpson's diversity index for all productive rearrangements in each sample. A public TCRβ was defined as a productive TCRβ shared by at least 2 individuals from our cohort by matching the CDR3β, Vβ,

and Jβ segments at the amino acid level. Additionally, a rank–size frequency distribution analysis was used to assess the differences in TCR sharing between the LS Survivors, LS Previvors, and controls, which in different subjects follows a power-law distribution, represented by the equation $f(x) = kx^{-\alpha}$[17,18]. A larger power exponent ($\alpha$) indicates that the population shares more public TCRs. Moreover, the Morisita's index was used to assess levels of similarity among samples by considering the relative frequencies of TCRβ along with the presence of specific TCRβs in more than one sample. Overlap between tissue and blood samples was assessed by matching the CDR3β, Vβ, and Jβ segments at the amino acid level.

### TCRβ annotation

Assessment of viral and autoimmune conditions-specificity for all public TCRβs was performed by annotating them against the Pathogens and Autoimmune categories from the McPAS-TCR database[19], based on exact matches of the CDR3β, J, and V segments at the amino acid level. For the assessment of MMRd CRC-specific public TCRβs, nine MMRd colorectal adenocarcinomas, from which TCR sequencing was performed using the same TCRβ ImmunoSEQ assay from Adaptive Biotechnologies, were pooled into a single MMRd CRC-specific TCRβ dataset. This dataset included three adenocarcinomas from our tissue cohort (Table S2), three MMRd colorectal adenocarcinomas obtained at MDACC from individuals not included in our PBMCs/Blood sample cohort, and three MMRd colorectal primary tumors from a publicly available dataset accessible through the immuneACCESS platform (https://doi.org/10.21417/B7WW2C) and previously published[20]. Public TCRβs from the PBMC/Blood sample cohort were annotated against the MMRd CRC-specific TCRβs pool by exactly matching the CDR3β, J, and V segments at the amino acid level.

### Phenotype association Analysis

The TCRβ sequence and its abundance in the peripheral blood were used as input features in the classifier to differentiate between LS carriers (previvors N = 130, and survivors N = 102, total N = 232) and controls (N = 45), between LS previvors and controls, and between LS survivors and previvors (Fig. 5A). To balance the number of control samples, we included 197 healthy (control) samples from a public dataset (Supplementary Data 3, https://www.synapse.org/Synapse:syn61987835/). We then randomly selected 20% of the samples as the validation set using the R 'sample' function (seed = 1234) and used the remaining 80% as the training/testing set to optimize the parameters (Supplementary Data 2). We considered the following hyperparameters: 1. TCR region: we considered the most optimal region to use CDR3aa only, CDR3aa plus J segments, CDR3aa plus V segments, or CDR3aa plus both J and V segments; 2. Positive presence ratio cutoff: The minimum percentage of positive-cohort individuals in which a TCRβ clone must be present; 3. Fisher's test cutoff: The significance threshold for a TCRβ clone's enrichment in the positive vs negative group in Fisher's test; 4. Wilcoxon test cutoff: The significance threshold for a TCRβ clone's increased expansion in the positive vs negative group in Wilcoxon test. We performed a grid search over combinations of these four parameters including TCR region (CDR3aa, CDR3aa+J, CDR3aa+V, CDR3aa+J + V), positive presence ratio cutoff (10%, 50%, 75%), Fisher's test cutoff ($10^{-1}, 10^{-2}, 10^{-4}, 10^{-10}, 10^{-15}$), and Wilcoxon test cutoff ($10^{-1}, 10^{-2}, 10^{-4}, 10^{-10}, 10^{-15}$, Fig. 5A) on the training set using the Leave-One-Out Cross Validation (LOOCV). After signature TCRs were identified, their mean expansion within a sample was used to predict the sample's phenotype using Gaussian Naïve Bayes classifiers. We assumed that the arcsine square root transformed mean proportions of these signature TCRs in each sample followed a Gaussian distribution with parameters (mean and standard deviation) estimated from both the positive and negative groups. Based on these distributions, we estimated the likelihood of a sample belonging to the positive group using the probability density functions of both the

positive and negative groups. We analyzed the trade-off between the true positive rate (TPR) and false positive rate (FPR) across various probability thresholds and reported AUROCs (area under the receiver operating characteristic curve) for the classifiers. Optimal parameters were selected based on the AUROC values from the LOOCV results of the training/testing set. In cases where AUROC values were tied at the second decimal place, we prioritized models with shorter TCR regions and smaller signature sets. The signature TCRs identified from the training/testing set using the optimal hyperparameters were tested on the 20% held-out validation set to assess the potential overfitting.

## In vitro validation of TCRs viral and neoAg specificity

As part of the TCRβ annotation process, we identified a subset of TCRβ sequences within our circulating public TCRβ repertoire, which were previously reported in the McPAS TCR database as specific to viral antigens. Among these viral antigens, we focused on an Epstein Barr Virus (EBV) antigenic peptide (BMLF1, amino acid sequence: GLCTLVAML), since we detected that several public TCRβ sequences from the blood samples of survivors, previvors, and controls showed specificity towards this BMLF1 peptide. To validate our annotation results in silico, we utilized the BMLF1 peptide in conjunction with a tetramer to stimulate PBMCs from a healthy human donor, thereby obtaining BMLF1-specific CD8+ T-cells for subsequent single-cell TCR sequencing. We then compared the TCRβ sequences obtained from these cells to our annotated public TCRβ repertoire to assess the degree of overlap and validate the specificity of the identified TCRβ sequences.

In detail, healthy donor PBMCs were cultured in 12-well plates ($1.5 \times 10^6$/well) using R10 media (RPMI 1640 with L-glutamine (Cat #10040CV, Corning), 10% heat-inactivated FBS (Cat# SH30070.03, HyClone), 10 mM Hepes buffer (Cat# 25060-CI, Corning), and 1X pen/strep (Cat#30002CI, Corning) supplemented with recombinant human IL-7 (R&D Systems Biotech, 330U/ml). PBMCs were stimulated with the BMLF1 (GLCTLVAML at 6 µg/ml). On days 3 and 7 of culture, cells were fed with R10 media in the presence of IL-2 (R&D Systems Biotech, 20 U/mL). On day 10, cells were boosted with R10 media in the presence of high dose IL-2 (1000U/ml). On day 12, cells were harvested, and CD3+ pan T-cells were isolated for subsequent flow cytometry analysis. Unstimulated T-cells (>96% purity) were isolated from PBMCs of HLA-A*02:01-positive healthy human donors using the Pan T-cell Isolation kit from Miltenyi Biotec (Bergisch Gladbach, Germany). Briefly, non-T-cells were depleted from PBMC using biotin-conjugated Abs against CD14, CD16, CD19, CD36, CD56, and CD123, followed by glycophorin A anti-biotin-labeled magnetic beads and LS columns. To generate peptide-loaded tetramers, we used Flex-T™ HLA-A*02:01 UVX monomers (Biolegend) according to the manufacturer's instructions. In brief, 20 µL of the HLA-A*02:01 UVX monomer was mixed with 20 µL of the respective viral peptide diluted to 400 µM in PBS. Then, the mixture was UV irradiated (366 nm) for 30 min, followed by incubation for an additional 30 min at 37 °C in the dark. The irradiated solution was mixed with APC-streptavidin and PE-streptavidin (Biolegend) to generate the tetramers. Flex-T™ monomers were assembled with two different streptavidin conjugates (APC and PE) in separate reactions to ensure the highest specificity. Finally, we added D-biotin to the tetramer solution to block any remaining free APC and PE-streptavidin. The tetramers were always prepared the day before the staining and left at 4 °C in the dark overnight. For each sample, the protein kinase inhibitor Dasatinib (Cayman Chemicals) was added to the culture medium to a final concentration of 50 nM to enhance the binding of fluorochrome-conjugated peptide-major histocompatibility complex (pMHC) tetramers and the cells were incubated for 1 hour at 37 °C. Then expanded pan T-cells were suspended in $Ca^{2+}/Mg^{2+}$ free Phosphate Buffered Saline (PBS), supplemented with 0.5% bovine serum albumin (BSA) (wash buffer), and were stained with R-phycoerythrin (PE) and Allophycocyanin (APC)- labeled pMHC-tetramer and Peridinin-Chlorophyll-Protein (PerCP) mouse anti-human CD8 antibody (Cat no. 347314, BD Biosciences, San Jose, CA, USA) to determine the number of viral peptide-specific CD8+ T-cells. Cells were incubated for tetramer staining for 30 min in the dark at 4 °C at the manufacturer's recommended concentrations. After the tetramer staining, cells were washed twice with 2 ml of wash buffer, centrifuged at 300 g for 5 min at 4 °C, resuspended on the residual volume, and incubated with the anti-CD8 antibody for 30 min on ice. Dead cells were excluded by Sytox Blue staining (1 mM, Molecular Probes, Carlsbad, CA, USA). Unstained pan T-cells were used to detect auto-fluorescence or background staining. Stained cells were analyzed and sorted using a CytoFLEX SRT Flow cytometer (Beckman Coulter, USA) under sterile conditions, and the results were analyzed by FlowJo Software version 10.8.1 (Tree Star, Inc., Ashland, OR, USA).

## In vitro validation of the signature TCRs specificity using RNF43 and MSH3 neoAgs

We selected RNF43 (with amino acid sequence: TQLARFFPI), one of the most recurrent peptides in LS colorectal cancers and pre-cancers with validated immunogenicity that has been identified in our previous work[1]. We used this RNF43_3 peptide in conjunction with a custom dextramer (Immudex) to stimulate PBMCs from 2 LS patients, thereby obtaining RNF43-specific T-cells. Subsequently, we performed single-cell RNA and single-cell TCR sequencing on these RNF43-dextramer-specific CD3+ and CD8+ T-cells. In parallel, a second recurrent neoAg peptide derived from a frameshift mutation in MSH3 (FLLALWECSL), previously described as a shared and immunogenic neoAg in LS carriers[8,22], was used to stain PBMCs from an additional LS carrier using an MSH3- specific HLA-A02:01 dextramer. We then compared the TCRβ sequences obtained from these cells to our LS-associated signature TCRβs and separately to our LS-survivor-associated signature TCRβs to assess overlap and validate the neoAg specificity of the identified TCRβ signature sequences.

In detail, PBMCs from LS patients carrying the HLA-A*02:01 allele were thawed and cultured in 12-well plates at a density of $1.5 \times 10^6$ cells/well in R10 medium (RPMI 1640 supplemented with L-glutamine (Catalog #10040CV, Corning, NY), 10% heat-inactivated FBS (Catalog #SH30070.03, HyClone, Cytiva, MA), 10 mM HEPES buffer (Catalog #25060-CI, Corning), and $1 \times$ Penicillin-Streptomycin (Catalog #30002CI, Corning). For some assays, this medium was further enriched with 1000 IU/ml human GM-CSF (Catalog #215-GM-050/CF, R&D Systems, MN), recombinant human IL-4 (Catalog #202-IL-050/CF, R&D Systems, MN), and human Flt-3L (Catalog #308-FKN-025, R&D Systems, MN), and incubated overnight (day 0). On day 1, PBMCs were stimulated with RNF43 or MSH3 peptides (1 µM) without viral peptides. Cultures were supplemented with human recombinant cytokines: IL-2 (10 IU/ml; Catalog #202-IL-050, R&D Systems, MN), IL-7 (10 ng/ml; Catalog #207-IL-025, R&D Systems, MN), and IL-15 (10 ng/ml; Catalog #200-15, PeproTech). After 24 h (day 2), cells were harvested and washed with PBS (without $Ca^{2+}/Mg^{2+}$) containing 0.5% BSA. For staining, cells were incubated with HLA-A*02:01 RNF43 (TQLARFFPI) and MSH3 (FLLALWECSL) dextramer-PE reagents (10 µl per 50 µl PBS containing 50% FBS; Immudex) for 30 min at room temperature. Following staining, cells were washed twice, centrifuged at 300 g for 5 min at 4 °C, and resuspended in residual buffer. Cells were then stained with anti-CD3-FITC (Catalog #561807, BD Biosciences, CA), anti-CD8-PerCP (Catalog #340693, BD Biosciences, CA), and 4-1BB-APC-Cy7 (Catalog #309830, BioLegend, CA) antibodies for 20 min on ice. Dead cells were excluded using Sytox Blue (1 mM, Molecular Probes, CA). Controls included unstained PBMCs for auto-fluorescence and HLA-unmatched healthy donor PBMCs as negative controls. Cells were analyzed and sorted on a CytoFLEX SRT flow cytometer (Beckman Coulter, CA), and data were processed using FlowJo v10.8.1 (Tree Star, OR).

### Cell preparation for single-cell profiling

To maximize cellular viability and recovery, samples were processed according to the 10x Genomics Demonstrated Protocol Cell Preparation Guide (CG00053; 10x Genomics, Pleasanton, CA, USA). Briefly, after removal of the supernatant, T-cells were gently washed twice in 1.5–3 ml of 0.04% BSA in PBS. Additionally, cells were passed through a 35μm strainer (BD Falcon® 5 ml Round-Bottom Tubes with Cell Strainer Cap, Corning, NY, USA) to eliminate cell clumps. Next, cells were stained with 0.4% Trypan blue and quantified and assessed for viability using the automated cell counting machine Countess 3 (Invitrogen, FL, US).

### 10x Genomics 5′ single cell RNA and V(D)J sequencing

Single-cell RNA-seq (scRNA) and single-cell TCR-seq (scTCR) analyses were performed using the 10x Genomics Single Cell Immune Profiling Solution V1.0 according to the manufacturer's protocols (10x Genomics V(D)J + 5′ Gene Expression). The scRNA libraries were sequenced on an Illumina HiSeq 3000 system using read lengths of 26 bp read 1, 8 bp i7 index, 98 bp read 2. The scTCR libraries were sequenced on an Illumina HiSeq 3000 using read lengths of 150 bp read 1, 8 bp i7 index, 150 bp read 2. Sequencing data were preprocessed separately using Cell Ranger 2.1.1, which included generating fastq files, aligning reads, and calculating gene-cell expression matrices. TCR reads were aligned to the GRCh38 reference genome, and consensus TCR annotation was performed using the cellranger vdj program. Barcodes with a higher number of Unique Molecular Identifier (UMI) counts than simulated background were identified as cell barcodes. For each barcode, cellranger performed de novo assembly and identified productive contigs along with their corresponding CDR3 regions and V, D, J, C genes.

The single-cell gene expression results were processed using Cell Ranger v8 multi and aligned to the GRCh38 reference genome. The seq counts were then analyzed using the Seurat v5 pipeline. Cells with more than 5% mitochondrial gene content were filtered out. Data normalization was performed using the log-normalization method, and variable features were identified. Principal Component Analysis (PCA) was conducted, and the first 30 principal components (PCs) were used for subsequent analyses, including clustering and visualization with Uniform Manifold Approximation and Projection (UMAP). The top marker genes for each cluster were identified and ranked by the BH-adjusted P value using the Wilcoxon test, and cell types were manually annotated.

### Statistical analyses

PRISM8 was used to perform all statistical analyses. The Mann-Whitney test was used to assess statistical significance of the differences among LS survivors, LS previvors and controls in the context of clonality, diversity, Morisita's index, the occupation of the repertoire space of hyper-expanded, large, medium, and small TCRβs, the gender of the individuals, the institution from which the samples were collected and the MMR gene with the germline pathogenic variant. A Chi-square test was used to assess the statistical significance of the differences among the percentage of TCRβs from each disease-specific pool that annotate to the LS cohort of survivors and previvors, and controls. Moreover, the power law distribution of the TCRs present in subjects was estimated using a linear model. This was done after log-transforming the percentage of positive subjects and the percentile rank of the TCRs ($\log10(y) = \log10(k) + \log(C) *\alpha$). Additionally, to assess the effects of the LS disease status (survivor, previvor, or control) in combination with age, the institution from which the samples were obtained, the gender of the individuals, the MMR gene with the germline pathogenic variant and the input material (blood vs PBMCs), on the Simpson clonality index (dependent variable), a generalized linear model for gamma distributed data was applied using the Stats (version 3.6.2) package in R, given the gamma distribution of the Simpson clonality index variable. In more detail, the Simpson Clonality index was

established as the response variable, and disease status, the institution from which the samples were collected, the gender, the MMR gene carrying the germline pathogenic variant, the input material (PBMCs or Whole blood), as well as the age of the individuals were used as the predictor variables. To infer the significance of correlation between the individual's age and the Simpson clonality index, the non-parametric Spearman's rank correlation coefficient was used. For every test, significance was defined by a $P$ value $< 0.05$.

## Required References

### Reporting summary

Further information on research design is available in the Nature Portfolio Reporting Summary linked to this article.

## Data availability

The TCRseq data generated in this study are publicly available in Zenodo (https://zenodo.org/records/13141052) and the scRNA-seq data generated in this study are publicly available in GEO under accession number GSE289646 Source data are provided with this paper.

## Code availability

The code used in this study is publicly available at https://github.com/Vilarlab-MDACC/LS_TCR.

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

## Acknowledgements

We thank the patients and their families for their participation. We thank the staff of the CPRIT Single Cell Genomics Core at MD Anderson Cancer Center for the assistance with single-cell TCR and RNA sequencing and the staff of the Advanced Technology Genomics Core and Cancer Genomics Laboratory at MD Anderson Cancer Center for their initial assistance in sequencing libraries generated from tissues and blood samples. We would like to acknowledge the support from the University of Kansas Medical Center's Biospecimen Repository Core Facility staff including Maura Kluthe, Alex Webster, Eric Johnson, and Lauren DiMartino for identifying, collecting, and processing human specimens; the support from Araceli Garcia Gonzalez, Jacklyn Thompson and Pragya Mishra for their assistance in identifying, consenting, collecting and processing human specimens from the MD Anderson cohort. The authors are grateful to Karen Colbert for her editorial assistance in the preparation of the manuscript.

## Author contributions

N.D. conducted the bioinformatic pipelines for TCR classifier development. F.D. performed single-cell sequencing and immunological assays. A.M.B. initiated the study, performed bulk TCR sequencing and bioinformatic analyses, and drafted the initial manuscript. N.D., F.D., and K.M.S. contributed to data analysis, manuscript review, editing, and finalization. M.W.T. performed pathological interpretation of tissue section specimens. L.R.U., S.T., L.R., P.M.L., Y.N.Y., and E.V. provided clinical samples and associated clinical information. F.M., M.P., and G.C. critically reviewed the manuscript and provided clinical samples from Spain. A.B. provided clinical samples from Kansas. S.K., P.S., G.A.L., and A.R. provided critical input on study design. E.V. led conceptualization, supervised the study, secured funding, and contributed to data analysis, manuscript review, and editing. E.V. had full access to all the data in the study and takes responsibility for the integrity of the data and the accuracy of the data analysis.

## Competing interests

Eduardo Vilar (EV) had a consulting or advisory role with Janssen Research and Development, Recursion Pharma, Nouscom, Abbvie, Moderna, Permanence Bio and Parabilis. EV has received research support from Janssen Research and Development and Nouscom. EV has

equity in Permanence Bio. The remaining authors declare no competing interests.
