## [Transparent Peer Review file · Nature Communications]

Genomic Analysis of T cell Receptors Provides Novel Insights of Cancer Immunology in Lynch Syndrome Carriers

Corresponding Author: Professor Eduardo Vilar

Version 0:

Reviewer comments:

Reviewer #1

(Remarks to the Author)

The reviewer acknowledges that the analysis is novel, as this type of investigation has not previously been conducted in the context of Lynch Syndrome. The manuscript is well-written, and the topic serves as a logical extension of the authors' prior work.

However, the reviewer has several concerns that limit the overall impact of the study:

1- The correlation between TCR repertoire in tissue and blood is weakened by the inclusion of tissue samples that appear to lack proper molecular characterization. As stated in the manuscript: "we also analyzed 14 colorectal tissue samples obtained from polyps or cancers in the lower gastrointestinal tract of LS individuals from whom we obtained PBMCs or blood" and "14 colorectal lesions from LS carriers collected during their routine surveillance colonoscopy."

However, not all colorectal lesions detected during surveillance colonoscopy in Lynch Syndrome (LS) patients are mismatch repair deficient (MMRd). Including these samples in the analysis without confirming MMR status post-collection is problematic (PMID: 39660603, 37262391). The manuscript does not specify whether the tissues were tested for MMR deficiency or if the molecular alterations aligned with the individual's germline variant. This lack of validation represents a critical limitation in the study design.

2- The stated aim of the journal is to "represent important advances of significance to specialists within each field." In their manuscript, the authors acknowledge: "However, our algorithm produced only moderate performance in distinguishing LS previvors from survivors, and the number of markers identified was limited." While the study appears sufficiently powered to differentiate between cases and controls, this alone does not constitute a meaningful advancement, as germline testing is already available to identify unaffected individuals.

The authors do highlight a potentially valuable clinical utility, noting that "our LS-associated and LS previvors-associated signatures could offer a solution in instances in which individuals meet clinical criteria for LS (e.g., Amsterdam criteria) but have negative germline testing for pathogenic variants in MMR genes, thus leaving these cases unresolved." However, this specific application was not evaluated in the current study.

Additionally, the authors acknowledge: "Further analysis with a larger LS survivor cohort and diverse neoAg is necessary for deeper insights". The authors performed an in-vitro validation based on testing one specific neoAg (RNF43) in cells from one LS. How representative is this experiment of the overall population of neoAg among LS patients?

Given these limitations, the reviewer concludes that currently, the findings do not represent a significant advance likely to impact specialists within the field.

Reviewer #2

(Remarks to the Author)

This study provides a genomic analysis of T-cell receptors that provides new insights into cancer immunology in Lynch syndrome carriers. While very interesting, there are several concerns that should be discussed.

Study design: deaths should be included.

Small sample of colorectal cancer and precancerous lesions.

Small number of healthy subjects.

RESULTS:

Why did “cancer status” and “age” mainly have a significant impact on TCR clonality?

It is appreciable that up to 41% of the TCR β clones that were most extended in colorectal cancer/precancerous lesions were detectable in the blood of LS carriers. However, the sample was small, and a maximum of 41% may not be a biomarker. Conclusion: Can TCR β clones be easily collected from blood as patient samples in the future?

Reviewer #3

(Remarks to the Author)

The manuscript by Bolivar et al. describes a large-scale analysis of the T cell receptor repertoire (as measured by the TCR-beta sequence) in lynch syndrome (LS) carriers. From a cohort of 102 LS survivors, 130 LS previvors and 45 controls, the authors make a number of interesting observations, including the level of clonotype expansion and diversity in the 3 groups, the degree of TCR overlap between groups, and the extent of detectability of expanded tumor T cell clones in the blood. They then evaluate the potential utility of TCRs as biomarkers of disease by training prediction models to distinguish between LS carriers and controls and survivors from previvors.

In general, this is an interesting study and a very valuable dataset to study TCR repertoires. However, some of the conclusions are not sufficiently supported by the data, and a number of aspects should be clarified as detailed below.

Major comments:

1. Reporting performance of the classifiers on the training set is misleading. Training and testing machine learning models on the same data, especially when the parameter space is large (e.g. results in Figure 5 and 6), leads to overfitting and to unrealistic AUC values of 0.99. This is clear from the LOO experiments, where we see that by simply removing one of the patients at a time, the performance drops drastically.
2. Leave-one-out experiments are less biased, but they still suffer from some level of overfitting. From the text, it appears that several hyperparameters (such as the threshold for TCR β inclusion, threshold for Fischer test, level of expansion and types of segments included in the model) are chosen from the full dataset – including the data used for testing. To avoid overfitting, the signatures of TCRs used for the model should only be chosen on each training set, without using information from the test set. Finally, without performance evaluation in a completely independent cohort, the strength of the evidence will remain weak. These limitations should be discussed and conclusions adapted accordingly.
3. Multiple factors other than disease contribute to TCR repertoire. Indeed, the authors evaluated gender, MMR gene, age and institution/batch as covariates, and found that age significantly explained TCR repertoire. Genetic background, prior infections, geographical region, and especially HLA type should be investigated carefully as likely confounders.
4. The analysis presented in Figure 4 shows that the TCRs in the PBMCs from this study have an overlap with TCRs previously reported in other studies and databases (associated with tumors, viruses or autoimmunity). As expected, a larger proportion of tumor-associated TCRs are seen compared to the viral and autoimmune pools. However, it seems that the same pattern is observed in controls, previvors and survivors in the three pools (Figure 4B), indicating that, contrary to the conclusions, these tumor-associated TCRs are not specific to survivors or previvors. Additionally, this plot should take into account the size of each group – there are less than half controls than the other two categories, and this imbalance mirrors the % TCR from each pool in dMMR, pathogens and autoimmune pool (Figure 4B).
5. A similar effect of imbalance in the number of subjects per group is seen in Figure 2A. Intersection of TCRs between any pair of subjects in the control group is bound to be combinatorially much lower simply because the control group is smaller.
6. Throughout the paper, the authors claim that identified TCRs are colorectal cancer specific, e.g. “shared TCRs between tissue and blood represent an immune response against colorectal neoplasms”, “CRC and precancer-specific public TCRs are detectable in circulation”, “This study is the first to describe circulating TCR β s that recognize neoantigens derived from colorectal cancers and pre-cancers in LS”. However, this is not supported by the data. TCRs tumor reactivity is not shown, and the suggesting TCR β s enrichment within tumor-infiltrating TCRs only provides weak evidence and is inconclusive (as discussed in comment #4).
7. The authors conduct an experimental validation to test if some of the LS-associated TCR β s are reactive to a previously identified LS-associated MHC-I restricted epitope (recurrent neoantigen caused by frameshift mutation in RNF43). RNF43-(HLA-A * 02:01)-dextramer-positive CD3+ T cells are isolated from a single LS donor and subjected to scRNAseq + scTCRseq. 6 out of 1400 TCR β s were detected among the 1100 LS-associated TCR β s. Strangely, most RNF43-HLA-A * 02:01-dextramer-positive CD3+ T cells were CD4+ contrary to the expected. Although this unexpected results is discussed in the paper, the experiment seems to be inconclusive and should be replicated in other samples/donors.

Minor comments

1. The observations on the TCR repertoire overlap between colorectal lesions and blood among the top 100 expanded clones is interesting and informative. It would be useful to extend this analysis and calculate how the overlap changes depending on the N most expanded clones considered (e.g. top 10, top 50, top 100, top 1000...).

2. In this work, analysis of TCR repertoires is performed based on exact matches of CDR3 and V(D)J segments of the TCR beta chain. However, variants in VDJ segments or CDR3 positions that are not in contact with the pMHC are likely to be permissive in terms of recognizing the same antigen. It would be interesting to see whether there are TCR motifs in the repertoires, i.e. groups of TCRs with similar sequences that may recognize the same antigen. Several algorithms such as GLIPH ([10.1038/nature22976](https://doi.org/10.1038/nature22976)) or GIANA (<https://doi.org/10.1038/s41467-021-25006-7>) may be helpful in this kind of analysis.

3. Following on the previous point, including TCR similarity measures in the encoding of TCR sequences (as opposed to exact matches) may be beneficial towards the building of the classifiers. Did the authors consider this kind of approach?

4. the use of “one-tailed” Mann Whitney tests should be justified (eg page 9)

5. x-axis labels of Figure 8 C are misaligned

Reviewer #4

(Remarks to the Author)

Version 1:

Reviewer comments:

Reviewer #2

(Remarks to the Author)

You replied well, so the manuscript is suitable for publication.

Reviewer #3

(Remarks to the Author)

Thank you for considering our comments and suggestions. One important issue remains with regard to code sharing and reproducibility. While a GitHub repository has been provided, it is insufficient to reproduce the analyses and lacks essential information: 1) there is no code for training the classification model, including parameters and train/test design; 2) a “functions.R” file is loaded by the scripts but not provided in the repository; 3) the scripts are bare of comments and contain no information on data input formats; 4) there is information on R and package versions (e.g the “renv” system is recommended). In general, providing usable code and sufficiently detailed methodological descriptions is a minimum standard in bioinformatics. As it stands, the current methods fall below these standards, making it difficult for users or reviewers to reproduce and evaluate the work.

Reviewer #4

(Remarks to the Author)

Reviewer #5

(Remarks to the Author)

his manuscript investigates the important topic of the changes and evolution of T-cell receptors (TCRs) in Lynch Syndrome (LS) carriers in response to the common indels frequently observed in LS tumors. This is a novel and promising analysis that yields several fascinating findings. In my opinion, the most interesting aspect is the clearly observable phenotype in LS carriers even before a tumor becomes detectable—and perhaps even before it develops.

However, while the biological claim is important, the article focuses less on exploring the biological mechanisms and more on using the observation as a classifier. As a classifier, the results in the revised version are improved compared to the previous one but remain borderline. The inclusion of large control cohorts strengthens the analysis, yet the distinction between Previvors and Survivors is still not sharp enough. Moreover, regarding Q2, the large discrepancy in Figure 7D between the AUC scores of the training set (0.991) and the test set (0.732) suggests potential overfitting in the model.

Version 2:

Reviewer comments:

Reviewer #3

(Remarks to the Author)
No further comments

Reviewer #4

(Remarks to the Author)

Reviewer #5

(Remarks to the Author)

The authors addressed the moderate classification results, suggesting that they are due to the small sample size. While this is possible, the results may actually stem from the inherent similarity between survivors and previvors. If the latter is the case, even a larger dataset would fail to clearly distinguish between the two groups.

Although their analysis discusses these issues-and their main claim appears to focus on distinguishing previvors from controls-the explanation provided to the reviewers must still be clearly stated in the main text. Specifically, given these two possibilities, it remains unclear whether a larger sample size would resolve the issue.

Point by point responses - NCOMMS-25-07866-T

Reviewer #1 The reviewer acknowledges that the analysis is novel, as this type of investigation has not previously been conducted in the context of Lynch Syndrome. The manuscript is well-written, and the topic serves as a logical extension of the authors' prior work. However, the reviewer has several concerns that limit the overall impact of the study.

Question #1: The correlation between TCR repertoire in tissue and blood is weakened by the inclusion of tissue samples that appear to lack proper molecular characterization. However, not all colorectal lesions detected during surveillance colonoscopy in Lynch Syndrome (LS) patients are mismatch repair deficient (MMRd). Including these samples in the analysis without confirming MMR status post-collection is problematic (PMID: 39660603, 37262391). The manuscript does not specify whether the tissues were tested for MMR deficiency or if the molecular alterations aligned with the individual's germline variant. This lack of validation represents a critical limitation in the study design.

Response: We thank the Reviewer for highlighting this very important point and for the opportunity to provide additional information on the tissue samples. We have performed immunohistochemistry (IHC) of the MMR proteins based on the known germline mutations of these LS patients to assess the MMR status. Please note that despite our efforts to complete the IHC in all the samples, we did not have available residual material for all of them. As the Reviewer can see we had MMRd and MMRp samples with no clear distinctive patterns in the sharing of TCR clones between blood and tissue. We have now included these results in the *Results* section under Demographic and Clinical Characteristics of the LS Patient Cohort, which reads as follows: *“Among the 14 lesions, four were MMRd, and three were MMR-proficient (MMRp) by immunohistochemistry (IHC) analysis. In two lesions, the IHC results were intermediate due to ambiguous or weak staining. The remaining five samples had been exhausted for TCRseq, so MMR status was unavailable”* (lines 172-175) with corresponding details also added to the *Materials and Methods* section (lines 538–540). IHC results are also included in **Table S2**.

Question #2: The stated aim of the journal is to “represent important advances of significance to specialists within each field.” In their manuscript, the authors acknowledge: “However, our algorithm produced only moderate performance in distinguishing LS previvors from survivors, and the number of markers identified was limited.” While the study appears sufficiently powered to differentiate between cases and controls, this alone does not constitute a meaningful advancement, as germline testing is already available to identify unaffected individuals.

Response: We appreciate the Reviewer's thoughtful comment. We agree that germline testing remains the gold standard for diagnosing LS carriers and detecting their pathogenic germline MMR mutations. However, our findings provide a novel concept on how circulating TCR β repertoires reflect ongoing and historical cancer-associated (and pre-cancer) immune surveillance responses in LS carriers. In response to this Reviewer and Reviewer #3, and to ensure a more balanced comparison group, we incorporated a large number of healthy controls from a public dataset. In this context, we observed similar performance in distinguishing LS carriers from average-risk individuals, further supporting the potential of circulating TCR β signatures as complementary immunologic biomarkers of LS-associated cancer risk. We agree with Reviewer

#1 that proposing the use of TCR signatures to diagnose LS is not advantageous or even cost-efficient, but it is within our reach to utilize these signatures in the near future to identify previvor and survivor status, and to further refine cancer surveillance and screening recommendations in LS carriers. In agreement with Reviewer #1 statements, we acknowledge that our model showed only moderate accuracy in distinguishing LS survivors from previvors. As discussed in the manuscript, this is likely due to overlapping neoAg exposure between these groups, combined with the use of the full circulating TCR repertoire, which includes many non-tumor-related clonotypes. We also added within the manuscript: *‘isolating and profiling neoAg-specific T cells remains technically and logistically challenging to apply at scale, especially in large patient cohorts’* (see **lines 490-491**). Thus, while our classifier does not yet enable precise stratification between survivors and previvors, we believe the demonstration that tumor- and neoAg-associated public TCR β s are detectable in the circulation of LS carriers represents a useful step toward developing blood-based immune biomarkers. Following Reviewer #1’s comment, we have now introduced a clear statement in the revised manuscript to clarify and provide better context on the outcomes of our findings. The sentence reads as *‘However, and despite this limitation, our findings suggest that these circulating TCR β repertoires might prove to be potential TCR biomarkers for assessing immune surveillance status in LS carriers.’* (**Discussion, lines 477-479**).

Question #3: The authors do highlight a potentially valuable clinical utility, noting that “our LS-associated and LS previvors-associated signatures could offer a solution in instances in which individuals meet clinical criteria for LS (e.g., Amsterdam criteria) but have negative germline testing for pathogenic variants in MMR genes, thus leaving these cases unresolved.” However, this specific application was not evaluated in the current study.

Response: We thank the Reviewer for this observation and agree that this potential clinical application was not directly evaluated and explored in our current study. Our conclusion was not to suggest using TCR β signatures as a diagnostic tool at this moment without having them validated. The scope of the present study is exploratory; however, this data suggests that these signatures may offer complementary immune profiling information in cases where germline testing results are inconclusive (e.g., VUS). In fact, we have an ongoing project open in our group to reclassify VUS in MMR genes based on TCRseq profiles. To avoid misinterpretations of our statement, we have revised the text that now reads: *‘These TCR β repertoires have potential as immune biomarkers; however, further validation in independent cohorts is necessary to assess their clinical utility and tumor reactivity’* (see in **Discussion lines, 509-511**).

Question #4: Additionally, the authors acknowledge: “Further analysis with a larger LS survivor cohort and diverse neoAg is necessary for deeper insights”. The authors performed an in-vitro validation based on testing one specific neoAg (RNF43) in cells from one LS. How representative is this experiment of the overall population of neoAg among LS patients? Given these limitations, the reviewer concludes that currently, the findings do not represent a significant advance likely to impact specialists within the field.

Response. We thank the Reviewer for this insightful comment. We acknowledge that the *in vitro* validation initially focused on a single neoAg, RNF43, and we used samples from one LS carrier to test the proof of concept. However, the RNF43 neoAg was specifically selected because its recurrence and immunogenicity were observed among LS-associated colorectal neoplasia. This

has been demonstrated in our previous work (Bolivar *et al* 2024, Ref #21 PMID: 38244726), where RNF43 neoAg was ranked among the top shared neoepitopes across LS carriers. In addition, RNF43 has also been reported in microsatellite instability (MSI) tumors more broadly and elicited robust CD8⁺ T cell responses across diverse HLA backgrounds (see Roudko, V. *et al.* 2020, Ref#8 PMID:33259803). To address the Reviewer's concern and further support the conclusion, we have now included new data from two additional neoAg-specific TCRs: (i) a second RNF43-specific TCR clones isolated from an independent LS patient; (ii) an MSH3-specific TCR, which is another recurrent and immunogenic neoAg described in both our group's prior studies and the literature (Bolivar *et al* 2024, Ref #21 PMID: 38244726, Roudko, V. *et al.* 2020, Ref#8 PMID:33259803). These TCRs were derived from LS patients within the same bulk sequencing cohort to maintain internal consistency. As described in the revised Results section (**lines 382 to 387**), we used a refined gating strategy in the follow-up experiments. In these validations, we detected 10.6% RNF43-specific CD8⁺ T cells in the second donor and 6.38% in the MSH3-specific sample (revised **Figure S6A, Figure S6B, lines 387-390**). Notably, upon intersecting the circulating LS-associated TCR β signature with TCRs from RNF43-dextramer-positive CD3⁺ and CD8⁺ T cells, we identified 80 overlapping CDR3 β sequences (**Table S14 and Figure 8C**). The LS-associated TCR β signature was derived from a classifier trained to distinguish all LS carriers (both survivors and previvors) from controls, thereby capturing the shared immune repertoire features of LS-associated immune surveillance. These shared TCR β s represent approximately 6.4% of the total unique CDR β sequences (n = 1249) obtained from the two RNF43-specific samples. From the MSH3-specific CD8⁺ T cell population, we identified 20 TCRs whose CDR3 β sequences overlapped with the circulating LS-associated TCR β signature, representing approximately 5.2% of the total unique CDR3 β sequences (n = 388) in that sample (**Table S14 and Figure 8D, lines 394-399**). In addition, we observed 783 CDR3 sequences overlapping with previously reported dMMR CRC-resident TCRs, including 8 that were expanded (Frequency > 0.05, **Figure 8E**) in the colorectal tumor samples (**lines 406-411**). Together, these new validation results further support the relevance of circulating neoAg-specific T cell receptors as biomarkers of immune recognition in LS.

Reviewer #2 This study provides a genomic analysis of T-cell receptors that provides new insights into cancer immunology in Lynch syndrome carriers. While very interesting, there are several concerns that should be discussed. Study design: deaths should be included. Small sample of colorectal cancer and precancerous lesions. Small number of healthy subjects.

Response: We thank the Reviewer for these important comments. Regarding clinical outcomes, available death information from all three contributing sites has now been incorporated into the clinical annotation table (**Table S16 "Vital Status", Column K**) for completeness and transparency. We added the text, which reads as follows: '*For the MDACC cohort, the median follow-up duration was 62 months calculated from the date of consent to the last recorded contact; three individuals were deceased at last follow-up. For the KUCC cohort, the median follow-up was approximately 72 months with one deceased individual. For the ICO cohort, the median follow-up duration was 138 months; however, vital status was unknown for six individuals. Across all three cohorts, aside from the four known deceased and the six with unknown status, all remaining participants across the three cohorts were alive at last follow-up*' (**Table S16**). These details have been added to the *Methods* section under 'Participants and samples' (**lines 521–533**).

Regarding the limited number of colorectal cancer and precancerous lesions included in our analysis, we fully agree and acknowledge the reviewer's concern that we had insufficient number of available tissue samples analyzed in this study. We agree that expanding the number of matched tissue samples in future studies will improve statistical power and generalizability. However, it is important to emphasize that all tissue samples were matched to individuals from our broader blood TCR repertoire cohort, which included 277 participants (102 LS survivors, 130 LS previvors, and 45 controls). The main focus of this study is to identify circulating TCR repertoires from LS cohorts and then test the overlap between circulating TCR β repertoires with tumor-infiltrating TCRs in available tissues. Notably, we observed that up to 41% of the top 100 most expanded TCRs in colorectal lesions were detectable in matched peripheral blood, and on average, 18% of these expanded tissue TCRs overlapped with the blood repertoire when collected concurrently. Moreover, several overlapping TCRs were public and detectable across multiple LS individuals, further strengthening the biological relevance of these findings despite the modest number of tissues. We have now modified the sentence which reads as: '*Therefore, our findings suggest that these TCRs may be targeting antigens derived from the colorectal tissue and T cell responses are detectable both in the lesion microenvironment and also in circulation.*' (lines 243-245). This overlap supports the core hypothesis that precancerous and cancerous colorectal lesions in LS carriers are immunologically recognized, and that these TCR responses can be captured in circulation. However, given the logistical and ethical constraints of collecting fresh colorectal lesions from LS carriers undergoing routine surveillance, we believe our matched blood-tissue dataset provides valuable, novel insight into immune surveillance in LS.

Furthermore, Reviewer #1 and #3 also raised a similar point regarding the imbalance in the lower number of control subjects. We have addressed this issue, specifically focusing on the limited number of healthy individuals. We added 197 healthy controls from public datasets (<https://www.synapse.org/Synapse:syn61987835>) to compensate for the imbalance of negative control (lines 290-291). The detailed explanation is included in the '*Phenotype Association Analysis*' section under the *Methods*, lines 591-593 (Table S5).

Question #1: Why did “cancer status” and “age” mainly have a significant impact on TCR clonality?

Response: We appreciate the Reviewer's questions and agree that understanding the influence of clinical variables, including 'cancer status and age', on TCR clonality is essential for interpreting repertoire dynamics. Both cancer status and age are well-established factors known to affect the TCR repertoire. In fact, age has been consistently associated with reduced TCR diversity and increased clonality due to thymic involution and the accumulation of memory T cells from lifelong antigenic exposures. This age-related reduction in the TCR repertoire has been described across multiple studies [Britanova et al., *Nat Commun* (2020), 11:4046; Sun et al., *J Clin Invest* (2022), 132(17):e158122]. This association was reflected in our dataset, where we observed weak but consistent positive correlations between age and clonality across all groups: LS survivors (Spearman's $R = 0.31$), LS previvors ($R = 0.31$), and controls ($R = 0.22$), as shown in **Figure S1A** (lines 194-196). We have now incorporated the age-related literature and discussion into the Discussion section which reads as follows: '*In our own dataset, we also observed weak but consistent positive correlations between age and TCR clonality across all groups, including LS*

survivors, LS previvors and controls, and confirmed age as an independent factor associated with clonality through multivariable modeling' (lines 436-439).

Regarding cancer status, particularly in the context of microsatellite instability and neoAg burden as seen in LS-associated tumors, this factor can lead to the selective expansion of tumor-reactive T cell clones. This antigen-driven expansion results in increased clonality, which has been reported in both tissue and peripheral TCR repertoires from patients with solid tumors [Yossef R et al, *Cancer Cell* (2023), Vol. 41(12), 2154-2165; Mehra and Taylor, *Haematologica* (2024), 109(7), 2038-2040; Matsuda T et al., *Oncoimmunology* (2019), 8(6), e1588085; He J et al., *Cell Res* (2022), 32(6), 530-542]. In our study, LS survivors exhibited significantly more clonal TCR β repertoires compared to LS previvors and controls (**Figure S1D**), likely reflecting immune memory from prior tumor exposure. We also found that this effect was especially pronounced in females (median SCI: 0.03655 for LS survivors vs 0.01900 for LS previvors, $P = 0.00011$; **Figure S1B**), and that TCR clonality did not significantly differ by institution or mutated MMR gene (**Figure S1C**). To further confirm these associations, we performed a generalized linear model (GLM) analysis including disease status, institution, gender, mutant MMR gene and HLAs as covariates. Only age and disease status remained statistically associated with clonality (SCI), as shown in **Table S3 (lines 204-205)**. Thus, both aging and antigen exposure from cancer likely contribute to the observed effects on TCR clonality in our cohort.

Question #2: It is appreciable that up to 41% of the TCR β clones that were most extended in colorectal cancer/precancerous lesions were detectable in the blood of LS carriers. However, the sample was small, and a maximum of 41% may not be a biomarker.

Response: We thank the Reviewer for making this observation. We agree that the tissue sample size is limited, and we have now emphasized this limitation more clearly in the revised text of the *Discussion*. Nonetheless, we believe that the detectable overlap up to 41% of expanded tissue TCR β clonotypes also found in blood provides preliminary evidence of a circulating tumor-reactive T cell pool in LS carriers. This finding supports the feasibility of monitoring tumor-associated immune responses through blood-based profiling. We agree that the 41% overlap, while promising, may not yet fulfill the criteria for a clinically actionable biomarker. However, we view this observation as a biologically meaningful signal that warrants further investigation in a larger and longitudinally followed cohort. As described in our earlier work (PMID: 38244726) and now supported by our current data, recurrent neoAg-specific clones can circulate in the blood and may reflect durable immune memory. We have now included a sentence in Discussion which reads as '*However, and despite this limitation, our findings suggest that these circulating TCR β repertoires might prove to be potential TCR biomarkers for assessing immune surveillance status in LS carriers*' (lines, 477-479).

Question #3: Can TCR β clones be easily collected from blood as patient samples in the future?

Response: We thank the Reviewer for this important question. As noted in the *Introduction* section (lines 141-145) of the manuscript, TCR-based tests have already entered clinical use for example, in the detection of CMV-specific TCRs in immunocompromised individuals using platforms such as Adaptive Biotechnologies' immunoSEQ and T-Detect™. Although our study did not utilize these clinical platforms, their successful application supports the feasibility of developing similar blood-based immune monitoring tools in the context of Lynch Syndrome. We believe this

highlights the broader translational potential of circulating TCR β profiling in high-risk cancer predisposition syndromes.

Reviewer #3 The manuscript by Bolivar et al. describes a large-scale analysis of the T cell receptor repertoire (as measured by the TCR-beta sequence) in lynch syndrome (LS) carriers. From a cohort of 102 LS survivors, 130 LS previvors and 45 controls, the authors make a number of interesting observations, including the level of clonotype expansion and diversity in the 3 groups, the degree of TCR overlap between groups, and the extent of detectability of expanded tumor T cell clones in the blood. They then evaluate the potential utility of TCRs as biomarkers of disease by training prediction models to distinguish between LS carriers and controls and survivors from previvors. In general, this is an interesting study and a very valuable dataset to study TCR repertoires. However, some of the conclusions are not sufficiently supported by the data, and a number of aspects should be clarified as detailed below.

Question #1: Reporting performance of the classifiers on the training set is misleading. Training and testing machine learning models on the same data, especially when the parameter space is large (e.g. results in Figure 5 and 6), leads to overfitting and to unrealistic AUC values of 0.99. This is clear from the LOO experiments, where we see that by simply removing one of the patients at a time, the performance drops drastically.

Question #2: Leave-one-out experiments are less biased, but they still suffer from some level of overfitting. From the text, it appears that several hyperparameters (such as the threshold for TCR β inclusion, threshold for Fischer test, level of expansion and types of segments included in the model) are chosen from the full dataset – including the data used for testing. To avoid overfitting, the signatures of TCRs used for the model should only be chosen on each training set, without using information from the test set. Finally, without performance evaluation in a completely independent cohort, the strength of the evidence will remain weak. These limitations should be discussed and conclusions adapted accordingly.

Response to Question #1 and #2: We thank the Reviewer for both comments and apologize for providing misleading statements in the manuscript. To address these comments, we have now split our initial dataset and used a subset as a validation set (20% of the total initial cohort). For training and testing purposes, we now perform leave-one-out (LOO) validation solely to identify the optimal parameters for determining the signature TCRs on the training/testing set. We have revised the methods section and clarified our design and procedure in the revised version of the manuscript (**lines 291-297**). The detailed explanation is included in the 'Phenotype Association Analysis' section under the Methods, **lines 591–620 (Table S6)**.

Question #3: Multiple factors other than disease contribute to TCR repertoire. Indeed, the authors evaluated gender, MMR gene, age and institution/batch as covariates, and found that age significantly explained TCR repertoire. Genetic background, prior infections, geographical region, and especially HLA type should be investigated carefully as likely confounders.

Response: Thank you for the Reviewer's suggestion regarding the potential impact of HLA. We obtained HLA information for the samples from MDACC and incorporated it into **Table S16**. We also examined the top 10 most frequent HLA alleles in our cohort and found no significant

differences in clonality (Simpson's index) among individuals carrying these alleles. This updated analysis is included in **Table S4**.

Question #4: The analysis presented in Figure 4 shows that the TCRs in the PBMCs from this study have an overlap with TCRs previously reported in other studies and databases (associated with tumors, viruses or autoimmunity). As expected, a larger proportion of tumor-associated TCRs are seen compared to the viral and autoimmune pools. However, it seems that the same pattern is observed in controls, previvors and survivors in the three pools (Figure 4B), indicating that, contrary to the conclusions, these tumor-associated TCRs are not specific to survivors or previvors. Additionally, this plot should take into account the size of each group – there are less than half controls than the other two categories, and this imbalance mirrors the % TCR from each pool in dMMR, pathogens and autoimmune pool (Figure 4B).

Response: Thank you for your comments. We observe similar patterns among previvors and survivors across different pools, thus suggesting that LS carriers as a whole consistently show a higher number of annotated TCRs across all categories. In the current version of the manuscript, the percentage of TCRs has been normalized by the size of each category, thus making the comparisons meaningful. However, we currently do not have a clear biological explanation for why LS patients exhibit more annotated TCRs associated with autoimmune conditions and infections. Nonetheless, studies (Chiou *et al*, 2021, PMID: 33691136) have shown that T cells specific for viral antigens like EBV and influenza can be found at high frequencies in tumors and circulation even in individuals without virus-associated cancers. This has been attributed to TCR cross-reactivity between pathogen-derived and tumor or self-antigens, which may be more pronounced in immune surveillance contexts such as LS.

Question #5: A similar effect of imbalance in the number of subjects per group is seen in Figure 2A. Intersection of TCRs between any pair of subjects in the control group is bound to be combinatorially much lower simply because the control group is smaller.

Response: Thanks for your suggestion regarding the imbalance issue. To address this, we have included the Jaccard Index to normalize for population size differences and to better illustrate the overlap effect (**New Figure 2B**). The Jaccard index between Control and Previvor groups was lower than that between Control and Survivor or Previvor and Survivor groups.

Question #6: Throughout the paper, the authors claim that identified TCRs are colorectal cancer specific, e.g. “shared TCRs between tissue and blood represent an immune response against colorectal neoplasms”, “CRC and precancer-specific public TCRs are detectable in circulation”, “This study is the first to describe circulating TCR β s that recognize neoAgs derived from colorectal cancers and pre-cancers in LS”. However, this is not supported by the data. TCRs tumor reactivity is not shown, and the suggesting TCR β s enrichment within tumor-infiltrating TCRs only provides weak evidence and is inconclusive (as discussed in comment #4).

Response: We thank the Reviewer for this important comment and completely agree that definitive tumor-specific-TCR cannot be assigned to the majority of TCR β s identified in our study, as most were not functionally validated. Our interpretation was based on: (i) the significant enrichment of these TCRs in tumor-infiltrating lymphocytes compared to matched blood samples;

(ii) their expansion patterns; (iii) overlap with previously reported TCRs associated with MMRd colorectal cancer; (iv) our targeted validation of one RNF43-specific TCR. However, we acknowledge that these features are supportive but not conclusive of tumor specificity for TCR candidates. A future study is warranted to test the tumor reactivity for some of the TCR candidates. To address this concern, we have carefully revised the language throughout the manuscript to avoid any overstatements. Specifically, the sentence: *‘Given this high ratio of shared TCRs between the lesion’s microenvironment and PBMCs, we concluded that the immune response against colorectal neoplasms in LS patients is not confined to the lesion’s microenvironment but is also present in circulation, and that precancerous lesions are already being recognized by T-cells’* has been revised to: *‘Therefore, our findings suggest that these TCRs may be targeting antigens derived from the colorectal tissue and T cell responses are detectable both in the lesion microenvironment and also in circulation.’* (lines 243-245).

Similarly, the statement: *‘According to our findings, MMRd CRC and precancer-specific public TCRβs are detectable in circulation in LS carriers’* has been revised to: *‘Based on above findings that public TCRβs associated with MMRd CRC and infiltrating precancerous lesions are detectable in the circulation of LS carriers (combining both previvors and survivors), we aimed to’* (lines 267-273).

And finally, this statement: *‘This study is the first to describe circulating TCRβs that recognize neoAgs derived from colorectal cancers and pre-cancers in LS carriers, moving the field towards the identification of a blood based TCRβ cancer biomarker.’* has been revised to: *‘This study is the first to characterize circulating TCRβs in LS carriers associated with CRC and pre-cancers, thus representing a step towards the identification of potential blood-based TCRβ biomarkers for immune surveillance’* (lines 112-114).

Question #7: The authors conduct an experimental validation to test if some of the LS-associated TCRβs are reactive to a previously identified LS-associated MHC-I restricted epitope (recurrent neoAg caused by frameshift mutation in RNF43). RNF43-(HLA-A*02:01)-dextramer-positive CD3⁺ T cells are isolated from a single LS donor and subjected to scRNAseq + scTCRseq. 6 out of 1400 TCRβs were detected among the 1100 LS-associated TCRβs. Strangely, most RNF43-HLA-A*02:01-dextramer-positive CD3⁺ T cells were CD4⁺ contrary to the expected. Although this unexpected result is discussed in the paper, the experiment seems to be inconclusive and should be replicated in other samples/donors.

Response: We thank the Reviewer for this thoughtful comment. We agree that the initial RNF43-HLA-A*02:01 dextramer validation was conducted in a single LS donor and, therefore was limited in scope. In the previous version of the manuscript, we proposed multiple possible explanations for CD4⁺ dextramer⁺ T cells, including noncanonical or low-affinity binding of the dextramer in CD4⁺ T cells, underdetection of CD8⁺ T cells due to sorting limitations, and cross-presentation of the RNF43 peptide on MHC-II molecules. The latter explanation is further supported by our previous work (PMID: 38244726, **Figure S4A**), which showed predicted MHC-II binding overlap for the MHC-I restricted RNF43 neoAg peptide.

In addition, to further address this concern we have now expanded our validation to include: (i) performing additional dextramer staining to isolate RNF43-specific T cells from an independent LS patient, and (ii) conducting a parallel dextramer-based isolation of neoAg-specific T cells recognizing another frameshift mutation in MSH3, which is another recurrent and immunogenic neoAg described in both our group’s prior studies and prior literature (Bolivar *et al* 2024, Ref #21

PMID: 38244726, Roudko, V. *et al.* 2020, Ref#8 PMID:33259803). In these follow-up experiments, we refined our approach by specifically isolating dextramer-positive T cells with high CD8⁺ expression. This strategy allowed us to enrich for a more functionally pure population of neoAg-specific CD8⁺ T cells and helped avoid ambiguity arising from potential MHC-II presentation, which complicated interpretation in our earlier analysis. As a result of this improved specificity, the second RNF43 validation and the MSH3 validation both yielded CD8-enriched T cell populations (**Now in Figure S6A, Figure S6B, and Figure 8B**). Using this strategy, we identified 80 TCRβs from RNF43-specific CD8⁺ and CD4⁺ T cells, and 20 TCRβs from MSH3-specific CD8⁺ T cells, that overlapped with the circulating LS-associated TCRβ signature (Now in **Table S14, Figure 8C, and Figure 8D**). The data is now outlined in the revised text (**lines 387-401**). We believe that these additional validations and the updated classifier together provide stronger support for the presence of circulating, neoAg-specific CD8⁺ T cells in LS carriers.

Minor comments

Question #1: The observations on the TCR repertoire overlap between colorectal lesions and blood among the top 100 expanded clones is interesting and informative. It would be useful to extend this analysis and calculate how the overlap changes depending on the N most expanded clones considered (e.g. top 10, top 50, top 100, top 1000...).

Response: Thank you for your suggestion. We have incorporated this data into **Figure S7**, which illustrates the overlap between the top N tissue-derived TCRs and circulating TCRs and their corresponding expansion levels.

Question #2: In this work, analysis of TCR repertoires is performed based on exact matches of CDR3 and V(D)J segments of the TCR beta chain. However, variants in VDJ segments or CDR3 positions that are not in contact with the pMHC are likely to be permissive in terms of recognizing the same antigen. It would be interesting to see whether there are TCR motifs in the repertoires, i.e. groups of TCRs with similar sequences that may recognize the same antigen. Several algorithms such as GLIPH (10.1038/nature22976) or GIANA (https://doi.org/10.1038/s41467-021-25006-7) may be helpful in this kind of analysis.

Question #3: Following on the previous point, including TCR similarity measures in the encoding of TCR sequences (as opposed to exact matches) may be beneficial towards the building of the classifiers. Did the authors consider this kind of approach?

Response to Questions #2 and #3: Thank you for your comment. We have tested various methods, but they did not perform better than the relatively “naïve” exact matching approach. For reasons of explainability and interpretability, we chose to use the exact match method for the final results. However, in response to the Reviewer’s suggestions, we have now included results based on GLIPH, Levenshtein distance, and TCRdist (See **Table S7**). The corresponding findings have been incorporated into the revised manuscript between **lines 275 and 277**.

Question #4: the use of “one-tailed” Mann Whitney tests should be justified (eg page 9), and x-axis labels of Figure 8 C are misaligned

Response: Thank you for your comment. We have addressed this comment and corrected the Figure as requested (see **lines 180, 181, 187, and Figures 1 and 8**).

Point by point responses to Reviewers

Reviewer #2 (Remarks to the Author):

You replied well, so the manuscript is suitable for publication.

Response: *Thanks!*

Reviewer #3 (Remarks to the Author):

Thank you for considering our comments and suggestions. One important issue remains with regard to code sharing and reproducibility. While a GitHub repository has been provided, it is insufficient to reproduce the analyses and lacks essential information: 1) there is no code for training the classification model, including parameters and train/test design; 2) a “functions.R” file is loaded by the scripts but not provided in the repository; 3) the scripts are bare of comments and contain no information on data input formats; 4) there is information on R and package versions (e.g the “renv” system is recommended). In general, providing usable code and sufficiently detailed methodological descriptions is a minimum standard in bioinformatics. As it stands, the current methods fall below these standards, making it difficult for users or reviewers to reproduce and evaluate the work.

Response: *We appreciate your comments and suggestions to improve the transparency of our work and access to our methods. The requested code and the renv.lock file specifying the R environment have been added to ensure reproducibility. We have provided detailed comments and step-by-step instructions for executing it. All the new information can be found at https://github.com/Vilarlab-MDACC/LS_TCR*

Reviewer #4 (Remarks to the Author):

Response: *Thanks!*

Reviewer #5 (Remarks to the Author):

This manuscript investigates the important topic of the changes and evolution of T-cell receptors (TCRs) in Lynch Syndrome (LS) carriers in response to the common indels frequently observed in LS tumors. This is a novel and promising analysis that yields several fascinating findings. In my opinion, the most interesting aspect is the clearly observable phenotype in LS carriers even before a tumor becomes detectable—and perhaps even before it develops.

However, while the biological claim is important, the article focuses less on exploring the biological mechanisms and more on using the observation as a classifier. As a classifier, the results in the revised version are improved compared to the previous one but remain borderline. The inclusion of large control cohorts strengthens the analysis, yet the distinction between

Previvors and Survivors is still not sharp enough. Moreover, regarding Q2, the large discrepancy in Figure 7D between the AUC scores of the training set (0.991) and the test set (0.732) suggests potential overfitting in the model.

Response: *Thank you for your comments and suggestions. We agree that the classifier distinguishing LS previvors from LS survivors has suboptimal performance. This likely reflects both the limited sample size and, more importantly, the intrinsic biological similarity between these two groups of carriers (survivors and previvors), despite being in a different stage of their disease.*

In our previous study, we found that individuals with LS may carry a high burden of somatic mutations and corresponding neoantigens generated from indels even in pre-cancer samples in LS carriers (PMID: 38244726). As a result, LS previvors may already develop specific TCR repertoires toward these neoantigens in pre-cancer tissues, which persist in circulation even after tumors develops, thus making their repertoires similar to those in survivors. However, some of these TCR clones may have clonally expanded in tumors. The dynamics of the TCR repertoire evolution during the transition from normal to precancer and later to cancer are not well understood yet. One possibility is that novel neoantigens associated with driver mutations arise during the precancer-to-cancer transition, leading to additional tumor-specific TCRs. Another possibility is that the TCR repertoire remains largely unchanged, either because driver mutations are already present at the precancer stage or because cancer progresses through immune escape rather than the generation of new antigens. There is also the possibility that these two dynamics are both present in the LS population but occur in different patients.

In our dataset, we detected very few differentially abundant or incident TCRs between previvors and survivors, which is more consistent with the second possibility. However, due to the modest sample size, we avoided over-interpret this finding. Future exploration of this concept is warranted. To achieve this goal, larger cohorts and longitudinal sampling will be required to clarify the dynamics of the TCR repertoire during early stages of pre-cancer and cancer initiation. Therefore, it is beyond the scope of the current study.

*In response to Reviewer's #5 astute comment, we have now added new analysis that accounts for limited performance of classifier signatures between these two groups (see new **Figure S7**). In addition, we have modified the text and added clarifications to address these points in the revised manuscript (**lines 331-357 and 457-482**).*

REVIEWERS' COMMENTS

Reviewer #3 (Remarks to the Author):

No further comments

Thank you.

Reviewer #4 (Remarks to the Author):

Thank you.

Reviewer #5 (Remarks to the Author):

The authors addressed the moderate classification results, suggesting that they are due to the small sample size. While this is possible, the results may actually stem from the inherent similarity between survivors and previvors. If the latter is the case, even a larger dataset would fail to clearly distinguish between the two groups.

Although their analysis discusses these issues-and their main claim appears to focus on distinguishing previvors from controls-the explanation provided to the reviewers must still be clearly stated in the main text. Specifically, given these two possibilities, it remains unclear whether a larger sample size would resolve the issue.

*Thanks to Reviewer #5 for this important suggestion. We have revised the discussion (**Lines 465-486**) to address this point more explicitly. We are now incorporating our previous findings (**Reference #5**) which indicate that survivors may experience a high neoantigen load. We have clarified that the observed TCR overlap supports the hypothesis that the adaptive immune response is established early during the precancerous stage and then persists over time, thus rendering two groups immunologically indistinguishable. Thus, the classifier's performance may reflect this shared biological history rather than being solely attributable to limitations in sample size as pointed by Reviewer #5.*